# An essential role for an Fe-S cluster protein in the cytochrome *c* oxidase complex of *Toxoplasma* parasites

Rachel A. Leonard[1], Yuan Tian[2¤a], Feng Tan[2]*, Giel G. van Dooren[1]*, Jenni A. Hayward [1¤b]*

**1** Research School of Biology, Australian National University, Canberra, Australian Capital Territory, Australia, **2** Department of Parasitology, School of Basic Medical Sciences, Wenzhou Medical University, Wenzhou, Zhejiang China

¤a Current address: Department of Clinical Laboratory, Jiangyou People's Hospital, Sichuan Jiangyou, China
¤b Current address: Department of Biochemistry and Molecular Biology, Biomedicine Discovery Institute, Monash University, Clayton, Victoria, Australia
* tanfengsong@163.com (FT); giel.vandooren@anu.edu.au (GGvD); jenni.hayward@monash.edu (JAH)

## Abstract

The mitochondrial electron transport chain (ETC) of apicomplexan parasites differs considerably from the ETC of the animals that these parasites infect, and is the target of numerous anti-parasitic drugs. The cytochrome *c* oxidase complex (Complex IV) of the apicomplexan *Toxoplasma gondii* ETC is more than twice the mass and contains subunits not found in human Complex IV, including a 13 kDa protein termed *Tg*ApiCox13. *Tg*ApiCox13 is homologous to a human iron-sulfur (Fe-S) cluster-containing protein called the mitochondrial inner NEET protein (*Hs*MiNT) which is not a component of Complex IV in humans. Here, we establish that *Tg*ApiCox13 is a critical component of Complex IV in *T. gondii*, required for complex activity and stability. Furthermore, we demonstrate that *Tg*ApiCox13, like its human homolog, binds two Fe-S clusters. We show that the Fe-S clusters of *Tg*ApiCox13 are critical for ETC function, having an essential role in mediating Complex IV integrity. Our study provides the first functional characterisation of an Fe-S protein in Complex IV.

## Author summary

Complex IV is a canonical component of the electron transport chain (ETC) of eukaryotic mitochondria. Recent evidence indicates that considerable diversity exists in the protein composition of Complex IV in eukaryotes, although the functions of most novel Complex IV proteins remain a mystery. In this study, we characterize *Tg*ApiCox13, a novel Complex IV protein from the apicomplexan parasite *Toxoplasma gondii*. We demonstrate that *Tg*ApiCox13 is an iron-sulfur protein that is essential for parasite survival, functioning as a critical mediator of the activity and stability of Complex IV. We show that the Fe-S clusters of *Tg*ApiCox13 are critical for Complex IV function, playing a key role in mediating integrity of the complex. Our findings provide the first functional characterization of an

**Data Availability Statement:** All relevant data are within the Manuscript, Figures and Supporting Information files.

**Funding:** This work was supported by a National Health and Medical Research Council Ideas grant (GNT1182369) to GGvD, grants from the National Natural Science Foundation of China (81971962, 82172303) to FT, and an Australian Government Research Training Program Scholarship to JAH. The funders had no role in study design, data collection and analysis, decision to publish, or preparation of the manuscript.

**Competing interests:** The authors declare that no competing interests exist.

Fe-S cluster protein in Complex IV, highlighting the diversity that exists in a core complex of the mitochondrial ETC in eukaryotes.

## Introduction

Mitochondria house many core biochemical pathways and processes that are critical for cell survival, including the electron transport chain (ETC) [1,2]. In the mitochondrial matrix, substrates are catabolized through the action of dehydrogenases and oxidoreductases, with the electrons from these reactions donated to a small hydrophobic electron carrier in inner mitochondrial membrane called coenzyme Q (CoQ). CoQ exchanges electrons with the coenzyme Q:cytochrome *c* oxidoreductase complex (Complex III), which transfers them to the soluble intermembrane space protein cytochrome *c* (CytC). CytC acts as an electron shuttle between Complex III and cytochrome *c* oxidase (Complex IV). Complex IV transfers electrons from CytC to the terminal electron acceptor oxygen ($O_2$), producing water ($H_2O$). Electron transport through Complexes III and IV is coupled to the translocation of protons from the mitochondrial matrix into the intermembrane space, thereby generating a proton motive force across the inner mitochondrial membrane. This electrochemical gradient is ultimately utilised by ATP synthase (Complex V) to generate adenosine triphosphate (ATP), and is also critical for other mitochondrial processes such as protein import.

Electrons are passed within and between the ETC complexes via redox reactions mediated by copper, heme and iron-sulfur (Fe-S) cluster moieties bound by the catalytic protein subunits. For instance, the ETC Complex II subunit SdhB contains three Fe-S clusters, which transfer electrons derived from the oxidation of succinate to CoQ [3]. ETC Complex III also contains one Fe-S cluster conjugated to the Rieske subunit, which, along with the heme cofactors of the cytochrome *b* and cytochrome $c_1$ subunits, facilitates electron transfer reactions within Complex III [4,5].

Apicomplexans belong to a phylum of intracellular protozoan parasites that includes the causative agent of toxoplasmosis *(Toxoplasma gondii)*, an opportunistic parasite that infects approximately 2 billion people [6,7]. Apicomplexans belong to a larger group of eukaryotes called the alveolates, which include ciliates such as *Tetrahymena*, and photosynthetic eukaryotes such as dinoflagellates and chromerids. The mitochondria of these eukaryotes contain numerous differences from the well-characterised mitochondria of animals [8,9]. Of particular note, the ETC Complexes of alveolates harbor numerous proteins that do not occur in the equivalent complexes of other eukaryotes such as animals [8,10–13]. For example, Complex IV of apicomplexans is estimated to be between 460–600 kDa, larger than the ~200 kDa complex found in mammalian cells, and contains approximately a dozen unique proteins not present in mammalian Complex IV [8,11,12]. Complex IV of *Tetrahymena* is a 2.7 MDa dimer, containing 44 proteins not observed in other Complex IV structures [13,14]. Although differences in protein composition are clear, the functional roles of novel subunits in these complexes are poorly understood.

Among the alveolate-specific Complex IV proteins is a small 13 kDa protein that we have called ApiCox13 and has been termed COXFS in ciliates [8,12–14]. ApiCox13/COXFS has homology to the Fe-S cluster-containing human mitochondrial inner NEET protein (*Hs*MiNT; also known as CISD3 and Miner2). *Hs*MiNT is the least well-characterised member of a family of three human NEET proteins. The other NEET proteins include the outer mitochondrial membrane (OMM) protein mitoNEET (mNT, CISD1) and the endoplasmic reticulum/mitochondria-associated membrane protein NAF-1 (CISD2). MitoNEET and NAF-1

function as homodimers that contain one Fe-S cluster binding site per protomer, with each protein facing the cytosol and tethered to their respective membrane by an N-terminal transmembrane domain [15]. By contrast, *Hs*MiNT is a soluble mitochondrial matrix-localized monomeric protein with two Fe-S cluster-binding sites [16,17]. All three NEET proteins are able to transfer Fe-S clusters to other proteins and have roles in the homeostasis of mitochondrial iron and reactive oxygen species (ROS) [18,19]. Notably, *Hs*MiNT is not a part of human Complex IV [16,17].

ApiCox13/COXFS is, to our knowledge, the first example of a putative Fe-S cluster protein that is associated with Complex IV in mitochondria, although its functional role in the complex remains unknown. In this study, we set out to characterise the role of ApiCox13 in Complex IV of *T. gondii*. We demonstrate that *Tg*ApiCox13 is an essential component of Complex IV in these parasites, with the Fe-S clusters of *Tg*ApiCox13 being critical for Complex IV integrity and function. Our study represents the first functional characterization of an Fe-S cluster protein from Complex IV.

## Results

### *Tg*ApiCox13 is a component of Complex IV in *T. gondii*

*Tg*ApiCox13 was previously identified as a potential subunit of the mitochondrial ETC Complex IV in *T. gondii* parasites [8]. *Tg*ApiCox13 is annotated as a "Zinc finger CDGSH-type domain-containing protein" on ToxoDB (TGGT1_254030), and has homology to the human mitochondrial inner NEET (*Hs*MiNT) protein (NP_001129970.1, 38% identity, E-value $3e^{-11}$). Both *Tg*ApiCox13 and *Hs*MiNT are predicted to contain two Fe-S cluster binding sites, co-ordinated by cysteine and histidine residues found in two so-called CDGSH motifs found in the proteins [16] (Fig 1A). To determine whether the *Tg*ApiCox13 and *Hs*MiNT proteins contain any other distinguishing features, we performed bioinformatic analyses to predict the presence of putative mitochondrial targeting peptides and transmembrane domains. Analysis of the amino acid sequence of *Tg*ApiCox13 suggested that while it is not predicted to possess an N-terminal mitochondrial targeting peptide, it is predicted to have a C-terminal transmembrane domain (S1 Table). By comparison, *Hs*MiNT is predicted to have an N-terminal mitochondrial targeting peptide but not a transmembrane domain (S1 Table).

We analysed other eukaryotic genomes for the presence of NEET proteins that have two CDGSH motifs (S1 Table). Homologs of *Tg*ApiCox13 were identified in the apicomplexans *Plasmodium falciparum* and *Babesia bovis*, the chromerid *Vitrella brassicaformis*, the dinoflagellate *Symbiodinium microadriaticum*, the ciliate *Tetrahymena thermophila*, the amoebozoan *Dictyostelium discoideum*, the trypanosome parasite *Trypanosoma cruzi*, and the fruit fly *Drosophila melanogaster*. We did not identify homologs of *Tg*ApiCox13 in plants or dikarya fungi such as yeast (*Saccharomyces cerevisiae*), though they could be identified in chytrid fungi such as *Rhizoclosmatium globusum*.

Alignment of the *Tg*ApiCox13 and *Hs*MiNT amino acid sequences with their homologs revealed the presence of an N-terminal extension in the protein of humans, flies, ciliates and dinoflagellates, which was absent from *Tg*ApiCox13 and the equivalent protein in other apicomplexans and in chromerids (Fig 1A, green). Analysis of this N-terminal region predicts that these extensions act as mitochondrial targeting signals (S1 Table). The alignments also revealed the presence of a C-terminal extension in the proteins from apicomplexans, ciliates, dinoflagellates, *Dictyostelium* and *Rhizoclosmatium*, which is absent from the homologues in animals and trypanosomes (Fig 1A, magenta). These extensions were predicted to be transmembrane domains (S1 Table). Mapping this information onto a currently accepted

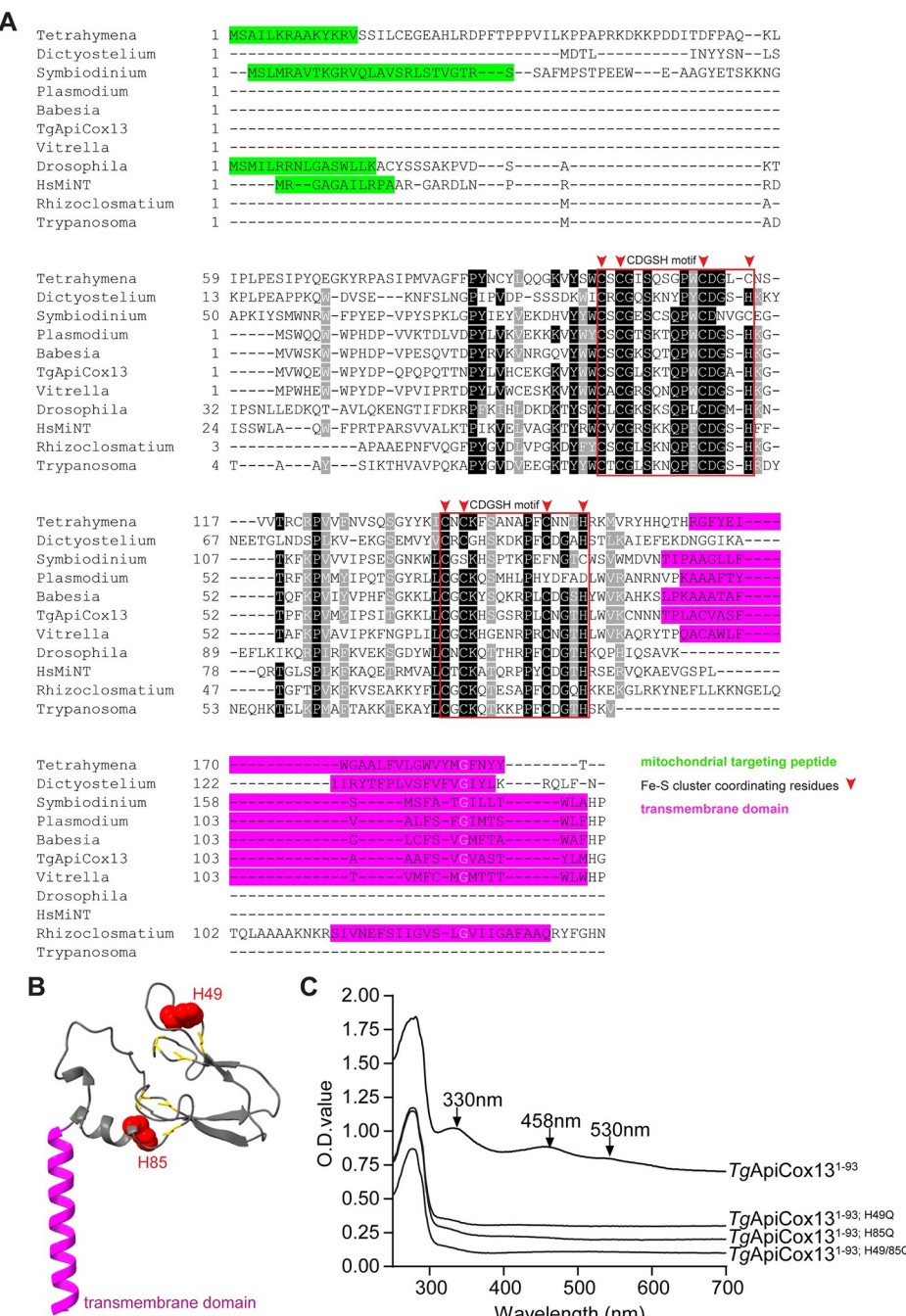

**Fig 1. *Tg*ApiCox13 is an Fe-S cluster binding protein. (A)** Multiple sequence alignment of *Tg*ApiCox13 and *Hs*MiNT with homologs from the apicomplexans *Plasmodium falciparum* and *Babesia bovis*, the trypanosome parasite *Trypanosoma cruzi*, the chromerid *Vitrella brassicaformis*, the dinoflagellate *Symbiodinium microadriaticum*, the ciliate *Tetrahymena thermophila*, the amoebozoan *Dictyostelium discoideum*, the fruit fly *Drosophila melanogaster* and the chytrid fungus *Rhizoclosmatium globosum*. Dark shading indicates amino acid identity in >70% of the sequences, and light shading indicates amino acid similarity in >70% of the sequences. Putative N-terminal mitochondrial targeting peptides are shown in green; putative C-terminal transmembrane domains are shown in magenta; CDGSH motifs are indicated in red boxes with conserved Fe-S cluster coordinating residues of the motifs depicted with red arrows. **(B)** The AlphaFold prediction of *Tg*ApiCox13 structure. The protein backbone is shown in grey, the putative transmembrane domain in magenta, the CDGSH motif cysteine residues as yellow sticks and histidine residues as red spheres. **(C)** UV-visible spectra of purified recombinant *Tg*ApiCox13¹⁻⁹³ and the Fe-S cluster binding site mutants *Tg*ApiCox13¹⁻⁹³; ᴴ⁴⁹Q, *Tg*ApiCox13¹⁻⁹³; ᴴ⁸⁵Q and *Tg*ApiCox13¹⁻⁹³; ᴴ⁴⁹/⁸⁵Q.

phylogenetic tree of eukaryotic evolution revealed that the common ancestor of alveolates likely had an ApiCox13 homolog with a C-terminal transmembrane domain (S1 Fig).

The key Fe-S binding residues of the CDGSH motifs of *Hs*MiNT, composed of three cysteines and one histidine, are highly conserved in *Tg*ApiCox13 and most other homologs (Fig 1A; red boxes and arrows). The AlphaFold prediction of *Tg*ApiCox13 structure reveals an overall fold very similar to that adopted by *Hs*MiNT (Fig 1B; [16,20,21]). However, AlphaFold predicts that *Tg*ApiCox13 contains a C-terminal α-helix that is not found in the *Hs*MiNT structure which, in combination with our bioinformatics analyses, we predict could function as a transmembrane helix (Fig 1B, magenta). Finally, AlphaFold predicts that the functional groups of the key Fe-S binding residues present in the two CDGSH motifs of *Tg*ApiCox13 are oriented in a similar manner to the equivalent residues of *Hs*MiNT (Fig 1B), which coordinate one Fe-S cluster each in the *Hs*MiNT structure [16].

To test whether *Tg*ApiCox13 is an Fe-S cluster binding protein, we expressed it in *E. coli* and assessed the purified recombinant protein by UV-visible absorption spectroscopy. In preliminary experiments, recombinant full-length *Tg*ApiCox13 had very low solubility. We therefore expressed *Tg*ApiCox13 lacking the predicted C-terminal transmembrane domain (*Tg*ApiCox13$^{1-93}$). *Tg*ApiCox13$^{1-93}$ was soluble and red in colour (S2A Fig), and exhibited three absorption peaks at 330, 458 and 530 nm (Fig 1C), which is characteristic of a [2Fe-2S] cluster binding protein [16,22]. According to our bioinformatic analyses, *Tg*ApiCox13 harbours two putative CDGSH motifs (Fig 1A). Substitution of one of the key Fe-S binding residues (histidine) in the CDGSH motifs of *Hs*MiNT with glutamine has been reported to abolish Fe-S cluster binding in *Hs*MiNT [22]. To test the ability of the two putative CDGSH motifs of *Tg*ApiCox13 to bind Fe-S clusters, we generated histidine to glutamine mutants in the two CDGSH motifs separately (*Tg*ApiCox13$^{1-93;\ H49Q}$, *Tg*ApiCox13$^{1-93;\ H85Q}$) and together (*Tg*ApiCox13$^{1-93;\ H49/85Q}$) and expressed them in *E. coli*. All three purified recombinant proteins lacked any visible colour (S2A Fig) even though they occurred in similar abundance to the *Tg*ApiCox13$^{1-93}$ protein (S2B Fig), and the absorption peaks attributed to the [2Fe-2S] clusters were absent (Fig 1C). Mutating the histidine residue of the CDGSH motif to cysteine has been demonstrated to increase the stability of [2Fe-2S] cluster binding in *Hs*MiNT [16]. We therefore generated histidine to cysteine mutants in the two CDGSH motifs separately (*Tg*ApiCox13$^{1-93;\ H49C}$, *Tg*ApiCox13$^{1-93;\ H85C}$) and together (*Tg*ApiCox13$^{1-93;\ H49/85C}$) and expressed them in *E. coli*. All three purified recombinant proteins were red in colour (S2A Fig) and displayed an absorption spectrum similar to *Tg*ApiCox13$^{WT}$ (S2C Fig).

Our spectroscopy analysis indicated that a mutation in one of the Fe-S cluster binding sites was indistinguishable from mutations in both sites. As a more sensitive measure to detect Fe-binding by *Tg*ApiCox13, we compared the iron content of the wild type *Tg*ApiCox13$^{1-93}$ protein to each of the CDGSH point mutants. We found that *Tg*ApiCox13$^{1-93}$ protein bound 1.12 ± 0.03 (mean ± SD, n = 3) mol of iron per mol of protein (S2D Fig). Assuming the presence of two [2Fe-2S] clusters in the protein (four Fe molecules per protein), we therefore estimate that ~25% of the recombinant *Tg*ApiCox13 proteins contain intact clusters in our experimental conditions. Mutation of histidines to cysteines in both CDGSH domains endowed *Tg*ApiCox13 with a slightly increased ability to bind to iron compared to the *Tg*ApiCox13$^{1-93}$ protein (S2D Fig). By contrast, the capacity of all histidine to glutamine mutants to bind iron was impaired (S2D Fig), with *Tg*ApiCox13$^{1-93;\ H49Q}$ and *Tg*ApiCox13$^{1-93;\ H85Q}$ binding ~0.42 ± 0.05 and 0.46 ± 0.01 mol of iron per mol of protein, respectively, and *Tg*ApiCox13$^{1-93;\ H49/85Q}$ binding 0.22 ± 0.02 mol of iron (mean ± SD, n = 3), approximately half the iron observed in the single point mutants. Taken together, these results support the hypothesis that *Tg*ApiCox13 is an Fe-S cluster binding protein containing two [2Fe-2S] clusters.

To investigate this protein in *T. gondii* parasites, we introduced a FLAG epitope tag into the 5' end of the open reading frame of the *Tg*ApiCox13 locus (S3A and S3B Fig). We did so in a *T. gondii* parasite line generated previously, in which the Complex IV protein *Tg*ApiCox25 had been HA tagged and the native promoter driving *Tg*ApiCox25 expression replaced with an anhydrotetracycline (ATc)-regulated promoter [8]. We termed the resulting line 'r*Tg*Api-Cox25-HA/FLAG-*Tg*ApiCox13'. Western blotting of proteins extracted from r*Tg*Api-Cox25-HA/FLAG-*Tg*ApiCox13 parasites using anti-FLAG antibodies detected a protein of ~18 kDa (Fig 2A), which is similar to the expected mass of the FLAG tag-containing*Tg*Api-Cox13 protein. Immunofluorescence assays performed using these parasites revealed that *Tg*ApiCox13 localised to the mitochondrion (Fig 2A), consistent with a recent study that demonstrated mitochondrial localisation of *Tg*ApiCox13 [23].

We next sought to determine whether *Tg*ApiCox13 is a part of Complex IV. To test this, we grew r*Tg*ApiCox25-HA/FLAG-*Tg*ApiCox13 parasites in the absence of ATc or in the presence of ATc for 1–3 days to deplete *Tg*ApiCox25 abundance, separated extracted proteins by SDS-PAGE, and performed western blotting. We found that, as predicted, *Tg*ApiCox25-HA was depleted following the addition of ATc (Fig 2B). By contrast, the abundances of the FLAG-*Tg*ApiCox13 protein and the *Tg*Tom40 loading control remained constant across the experiment (Fig 2B). To monitor Complex IV, we extracted proteins in 1% (v/v) Triton X-100 (TX-100) and separated these proteins by blue native (BN)-PAGE. Western blotting using anti-FLAG antibodies revealed that FLAG-*Tg*ApiCox13 is part of a ~600 kDa complex in the absence of ATc, approximately the same mass as the r*Tg*ApiCox25-HA complex when the blot was probed with anti-HA antibodies (Fig 2C). In the presence of ATc, the FLAG-*Tg*ApiCox13 complex breaks down into a smaller ~400 kDa complex, concomitant with the disappearance of the r*Tg*ApiCox25-HA complex (Fig 2C). Together, these data indicate that knockdown of *Tg*ApiCox25 influences formation of the *Tg*ApiCox13-containing complex, consistent with *Tg*ApiCox13 being in the same complex as *Tg*ApiCox25.

As a more direct test for whether *Tg*ApiCox13 is a Complex IV protein, we conducted a co-immunoprecipitation experiment. Immunoprecipitation of the known Complex IV protein *Tg*ApiCox25-HA using anti-HA antibody-conjugated beads co-purified FLAG-*Tg*ApiCox13, but not the unrelated mitochondrial protein *Tg*Tom40 (Fig 2D). Similarly, *Tg*ApiCox25-HA was co-purified upon immunoprecipitation of FLAG-*Tg*ApiCox13 using anti-FLAG antibody-conjugated beads but not *Tg*Tom40 (Fig 2D). We conclude that *Tg*ApiCox13 is a constituent protein of Complex IV in *T. gondii*.

## *Tg*ApiCox13 is important for *T. gondii* proliferation and ETC function

Having determined that *Tg*ApiCox13 is a part of ETC Complex IV, we next sought to elucidate the role and importance of *Tg*ApiCox13 in *T. gondii* parasites. To do this, we designed a CRISPR-Cas9 genome editing strategy to simultaneously replace the native promoter of *Tg*A-piCox13 with an ATc-regulatable promoter and incorporate a HA epitope tag onto the 5' end of the *Tg*ApiCox13 open reading frame (S3C and S3D Fig). We termed the resultant ATc-regulatable line 'rHA-*Tg*ApiCox13'. To determine the extent of *Tg*ApiCox13 knockdown upon the addition of ATc, we cultured rHA-*Tg*ApiCox13 parasites in the absence of ATc or in the presence of ATc for 1–3 days, then separated proteins by SDS-PAGE. Western blotting using anti-HA antibodies revealed that expression of *Tg*ApiCox13 was substantially decreased after 2 days on ATc (Fig 3A). To examine whether *Tg*ApiCox13 is important for parasite proliferation, we grew wild type (WT) or rHA-*Tg*ApiCox13 parasites in the absence or presence of ATc for 8 days and compared the size of plaques. In the presence of ATc, plaque size was severely impaired in rHA-*Tg*ApiCox13 but not WT parasites (Fig 3B). To ensure that the observed

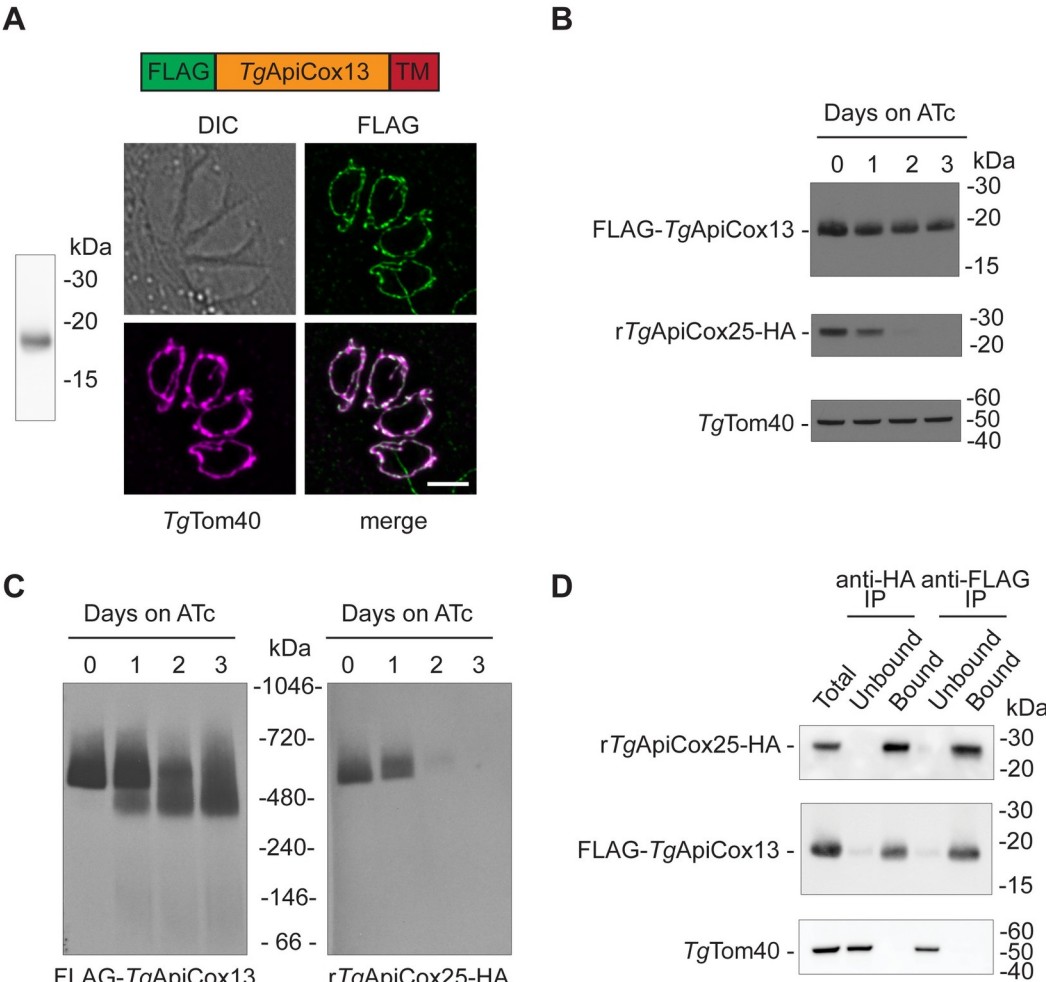

**Fig 2. *Tg*ApiCox13 interacts with *Tg*ApiCox25 in Complex IV of the mitochondrial ETC. (A)** Left, Western blot of proteins extracted from r*Tg*ApiCox25-HA/FLAG-*Tg*ApiCox13 *T. gondii* parasites, separated by SDS-PAGE and probed with anti-FLAG antibodies. Right, Immunofluorescence assay depicting four r*Tg*ApiCox25-HA/FLAG-*Tg*ApiCox13 *T. gondii* parasites in an intracellular vacuole. FLAG-*Tg*ApiCox13 was detected with anti-FLAG (green), and the mitochondrion with anti-*Tg*Tom40 (magenta), antibodies, with the overlap shown in the merge image. DIC, differential interference contrast; scale bar represents 2 μm. **(B-C)** Western blots of proteins extracted from r*Tg*ApiCox25-HA/FLAG-*Tg*ApiCox13 parasites cultured in the absence of ATc or in the presence of ATc for 1–3 days. **(B)** Samples were separated by SDS-PAGE, and probed with anti-HA, anti-FLAG and anti-*Tg*Tom40 (loading control) antibodies. **(C)** Samples were prepared in 1% (v/v) TX-100 detergent, separated by BN-PAGE, and detected with anti-FLAG (left) or anti-HA (right) antibodies. **(D)** Western blots of proteins extracted from r*Tg*ApiCox25-HA/FLAG-*Tg*ApiCox13 parasites, and subjected to immunoprecipitation using anti-HA (anti-HA IP) or anti-FLAG (anti-FLAG IP) antibody-coupled beads. Extracts include samples before immunoprecipitation (Total), samples that did not bind to the anti-HA or anti-FLAG beads (Unbound), and samples that bound to the anti-HA or anti-FLAG beads (Bound). Samples were separated by SDS-PAGE, and probed with anti-HA antibodies to detect *Tg*ApiCox25, anti-FLAG to detect *Tg*ApiCox13, and anti-TgTom40 as a control to detect an unrelated mitochondrial protein. Western blots shown are representative of three independent experiments, with matched BN-PAGE and SDS-PAGE samples prepared from the same experiment.

proliferation defect was specifically due to *Tg*ApiCox13 knockdown, we complemented the rHA-*Tg*ApiCox13 line with an additional copy of *Tg*ApiCox13 containing an N-terminal FLAG tag and constitutively expressed from an α-tubulin promoter (rHA-*Tg*ApiCox13/ cFLAG-*Tg*ApiCox13$^{WT}$). Complementation restored parasite proliferation in the presence of ATc (Fig 3B). Together, these data indicate that *Tg*ApiCox13 is important for parasite proliferation.

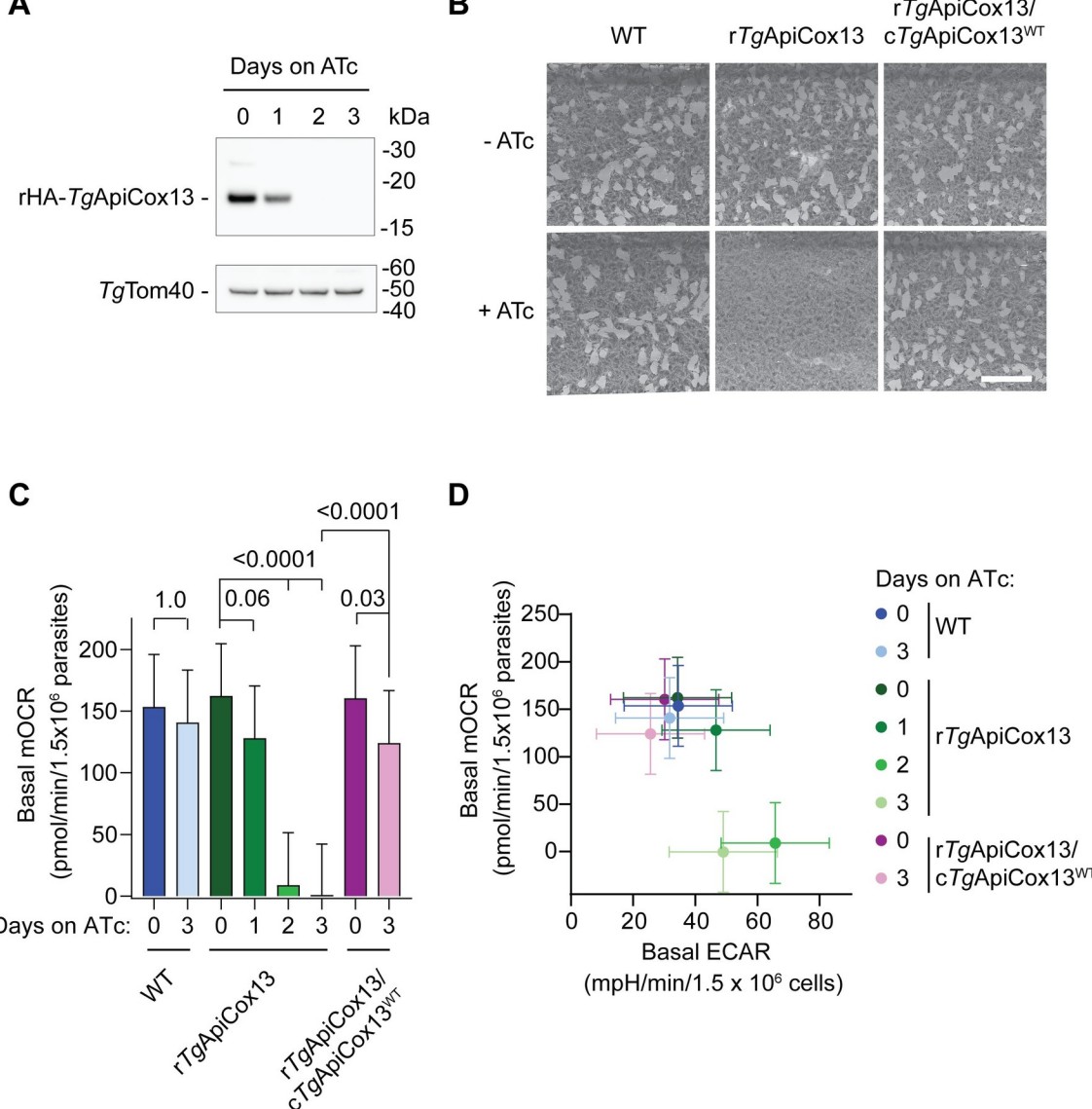

**Fig 3. *Tg*ApiCox13 is important for parasite proliferation and mitochondrial O₂ consumption. (A)** Western blot of proteins extracted from rHA-*Tg*ApiCox13 parasites cultured in the absence of ATc or in the presence of ATc for 1–3 days, separated by SDS-PAGE and detected using anti-HA antibodies (to detect the HA-*Tg*ApiCox13 protein) and anti-*Tg*Tom40 antibodies (loading control). **(B)** Plaque assays measuring proliferation of WT, rHA-*Tg*ApiCox13 and complemented rHA-*Tg*ApiCox13/cFLAG-*Tg*ApiCox13$^{WT}$ parasites cultured in the absence (top) or presence (bottom) of ATc for 7 days. Assays are from a single experiment and are representative of three independent experiments. Scale bar represents 1 cm. **(C)** Basal mitochondrial O₂ consumption rate (mOCR) of WT (blue), rHA-*Tg*ApiCox13 (green) and rHA-*Tg*ApiCox13/cFLAG-*Tg*ApiCox13$^{WT}$ (purple) parasites cultured in the absence of ATc or in the presence of ATc for 1–3 days. A linear mixed-effects model was fitted to the data and values depict the estimated marginal mean ± 95% CI of three independent experiments. ANOVA followed by Tukey's multiple pairwise comparisons test was performed with relevant *p* values shown. **(D)** Basal mOCR versus basal extracellular acidification rate (ECAR) of WT (blue), rHA-*Tg*ApiCox13 (green) and rHA-*Tg*ApiCox13/cFLAG-*Tg*ApiCox13$^{WT}$ (purple) parasites cultured in the absence of ATc or in the presence of ATc for 1–3 days. Data depict the estimated marginal mean mOCR and ECAR values ± 95% CI of three independent experiments.

We next sought to assess whether *Tg*ApiCox13 is important for ETC function. Since molecular oxygen acts as the terminal electron acceptor in the ETC, we measured parasite O₂ consumption using a Seahorse XFe96 extracellular flux analyzer to assess ETC functionality. We

grew WT, rHA-*Tg*ApiCox13 and rHA-*Tg*ApiCox13/cFLAG-*Tg*ApiCox13$^{WT}$ parasites in the absence of ATc or in the presence of ATc for 1–3 days and then measured their basal mitochondrial $O_2$ consumption rate (mOCR). The basal mOCR of rHA-*Tg*ApiCox13 parasites was significantly depleted upon *Tg*ApiCox13 knockdown (Fig 3C), while the mOCR of WT parasites was similar in the absence or presence of ATc (Fig 3C). Furthermore, complementation with constitutively-expressed cFLAG-*Tg*ApiCox13$^{WT}$ largely restored mOCR in the presence of ATc (Fig 3C).

In addition to measuring OCR, the Seahorse XFe96 extracellular flux analyzer concurrently measures the extracellular acidification rate (ECAR) of parasites, which we have used previously as a general measure of parasite metabolic activity and viability [10, 24–26]. Growth of rHA-*Tg*ApiCox13 parasites in the presence of ATc resulted in a slight increase in ECAR (Figs 3D and S4), indicating that the defect we observed in mOCR upon *Tg*ApiCox13 knockdown is not due to a general loss of parasite metabolism or parasite death. We speculate that the slight increase in ECAR, which we have also observed upon the depletion of other ETC proteins [8, 10], may reflect a compensatory mechanism, perhaps increased glycolytic flux, in response to decreased ETC function. Together, these data indicate that *Tg*ApiCox13 has an important role in the ETC of *T. gondii* parasites.

We and others have previously observed that although compromising ETC function by either the knockdown of ETC complex subunits [8, 10] or mitochondrial Fe-S cluster synthesis pathway proteins [23], or treatment with the Complex III inhibitor atovaquone [10], severely impairs *T. gondii* parasite proliferation, it is not fatal to parasites. We therefore wondered whether parasites also remain viable following knockdown of *Tg*ApiCox13. We pre-incubated r*Tg*ApiCox13 parasites in the presence of ATc for 3 days, then washed out the ATc and allowed parasites to proliferate in the absence of ATc for a further 9 days before measuring the extent of parasite proliferation by plaque assay. As controls in these experiments, we performed similar wash-outs in parasite lines generated previously, in which the cysteine desulfurase (*Tg*NFS1) that catalyzes the first step of the mitochondrial Fe-S cluster synthesis pathway [25], the Complex III subunit *Tg*QCR11 [10] or the Complex IV subunit *Tg*ApiCox25 [8] can be knocked down by the addition of ATc. We compared the resulting plaques to parasites that had not been pre-incubated in ATc or that were grown in ATc for all 12 days. We observed that all four parasite lines had a similar number of plaques after ATc washout compared to the no ATc control, while parasites grown in the presence of ATc for all 12 days underwent reduced proliferation (S5 Fig), with equivalent extents of inhibition in the ETC mutants and a greater degree of inhibition in the *Tg*NFS1 mutant. Interestingly, we observed minimal recovery in the ATc-regulatable r*Tg*NFS1-HA line following ATc removal, an equivalent level of recovery in the Complex IV mutant lines (rHA-*Tg*ApiCox13 and r*Tg*ApiCox25-HA), and a much greater level of recovery in the Complex III mutant line (r*Tg*QCR11-FLAG; S5 Fig). These results indicate that rHA-*Tg*ApiCox13 knockdown is reversible, which fits with an emerging narrative that parasites survive but do not thrive when the ETC is compromised. The data also indicate differences in the reversibility of depleting mitochondrial Fe-S cluster synthesis compared to ETC proteins, and between depleting Complex III compared to Complex IV proteins. The greater importance, and reduced reversibility following depletion, of mitochondrial Fe-S cluster synthesis for parasite proliferation compared to the ETC likely reflects the multiple essential cellular pathways that are dependent on this pathway for Fe-S clusters. The reasons for the differences in rescue following depletion of Complex III versus Complex IV proteins are less clear. This could reflect differences in the ability of the two complexes to reassemble upon the knockdown of key proteins, although it is also conceivable that Complex IV has additional functions beyond the ETC.

## *Tg*ApiCox13 is important for the activity and integrity of Complex IV

Since *Tg*ApiCox13 is a component of Complex IV and knockdown of *Tg*ApiCox13 led to decreased ETC activity, we hypothesised that *Tg*ApiCox13 is important specifically for Complex IV function. To test this, we utilized a modified Seahorse XFe96 flux analyzer assay established previously to more directly assess the functionality of Complex IV [10,24] (Fig 4A). We grew WT, rHA-*Tg*ApiCox13 and rHA-*Tg*ApiCox13/cFLAG-*Tg*ApiCox13[WT] parasites in the absence of ATc or in the presence of ATc for 1–3 days, then starved parasites for 1 hour to deplete endogenous substrates. We then permeabilized the parasite plasma membrane by

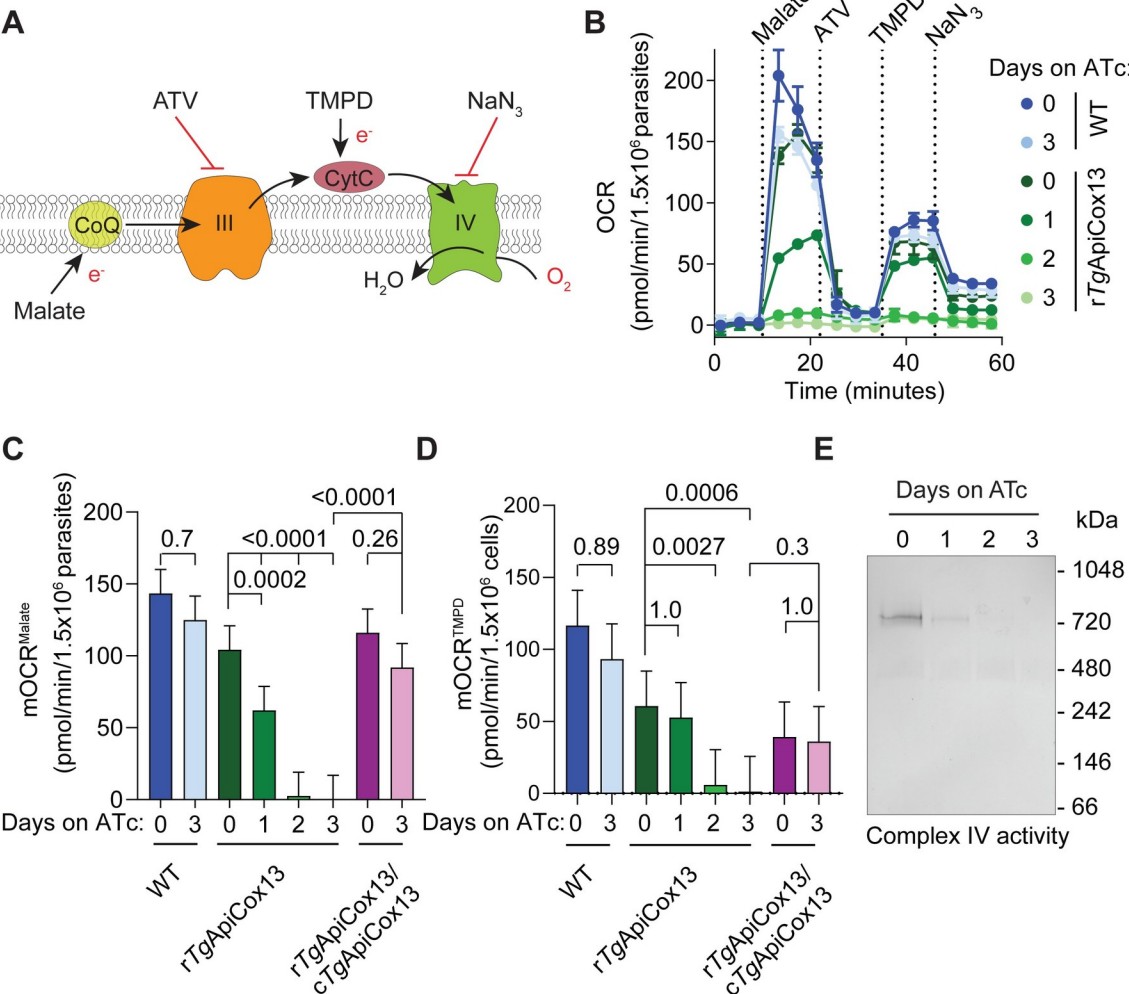

**Fig 4. *Tg*ApiCox13 is important for the function of Complex IV. (A)** Schematic of the assay probing ETC function, illustrating where substrates and inhibitors act on the ETC. Malate donates electrons (e⁻) to coenzyme Q (CoQ), increasing OCR when ETC is functional; atovaquone (ATV) inhibits Complex III (III), decreasing OCR; TMPD donates e⁻ to cytochrome *c* (CytC), increasing OCR when Complex IV (IV) is functional; sodium azide (NaN₃) inhibits Complex IV, decreasing OCR. **(B-D)** WT (blue), rHA-*Tg*ApiCox13 (green) and cFLAG-*Tg*ApiCox13/rHA-*Tg*ApiCox13 (purple) parasites were cultured in the absence of ATc or in the presence of ATc for 1–3 days and the assay was performed. **(B)** Representative trace depicting OCR over time. Data represent the mean ± SD of 3 technical replicates and are representative of 3 independent experiments. **(C-D)** Quantification of **(C)** malate-elicited mOCR (mOCR[Malate]) and **(D)** TMPD-elicited mOCR (mOCR[TMPD]). A linear mixed-effects model was fitted to the data and values depict the estimated marginal mean ± 95% CI of 3 independent experiments. ANOVA followed by Tukey's multiple pairwise comparisons test was performed with relevant *p* values shown. **(E)** In-gel Complex IV activity assay performed on samples extracted from an equal number of rHA-*Tg*ApiCox13 parasites cultured in the absence of ATc or in the presence of ATc for 1–3 days. The intensity of the band at ~720 kDa corresponds to the extent of Complex IV activity.

adding low concentrations of digitonin, enabling the subsequent uptake of ETC substrates by the parasites. Baseline mOCR readings taken before injection of substrate were close to zero (Fig 4B), indicating that parasites were not respiring and, therefore, that endogenous substrates had been successfully depleted. The substrate malate was then injected, resulting in an instantaneous increase in mOCR in WT parasites cultured in the absence or presence of ATc (blue) and in rHA-*Tg*ApiCox13 parasites cultured in the absence of ATc (dark green) (Fig 4B; quantified in Fig 4C). By contrast, the malate-elicited mOCR of rHA-*Tg*ApiCox13 parasites decreased over 1–3 days on ATc (Fig 4B and 4C). OCR was abolished by injection of the Complex III inhibitor atovaquone (Fig 4B), indicating that the malate-elicited mOCR is indeed occurring through the ETC. These results indicated that malate-dependent ETC function is decreased upon *Tg*ApiCox13 knockdown.

To determine whether Complex IV activity was specifically compromised, we then injected the CytC substrate N,N,N',N'-tetramethyl-p-phenylenediamine dihydrochloride (TMPD), which can bypass ETC defects upstream of CytC (*e.g.* in Complex III) to rescue OCR, but cannot rescue OCR if the defect occurs downstream of CytC (*i.e.* in Complex IV) (Fig 4A) [10,24]. While WT parasites were able to utilise TMPD as a substrate to stimulate $O_2$ consumption regardless of whether they were cultured in the presence or absence of ATc, we observed that $O_2$ consumption in rHA-*Tg*ApiCox13 parasites cultured for 2 or 3 days on ATc was not stimulated by TMPD (Fig 4B; quantified in Fig 4D). This is consistent with an ETC defect occurring downstream of CytC (*i.e.* in Complex IV) upon *Tg*ApiCox13 knockdown.

As a second measure of Complex IV activity, we grew rHA-*Tg*ApiCox13 parasites in the absence of ATc or in the presence of ATc for 1–3 days, extracted proteins in a 1% (v/v) TX-100-containing buffer, separated proteins by clear native-PAGE and performed an in-gel Complex IV assay as described previously [27]. Consistent with the oxygen consumption experiments, the in-gel activity of Complex IV was reduced upon *Tg*ApiCox13 knockdown (Fig 4E). Together, these results indicate that loss of *Tg*ApiCox13 leads to a specific defect in Complex IV activity.

We wondered whether the integrity of Complex IV was compromised by the loss of *Tg*ApiCox13. In yeast and other organisms, Complex IV subunits are grouped into three modules based on their association with the three core catalytic subunits Cox1, Cox2 or Cox3 [28]. While a structure of *T. gondii* Complex IV is currently lacking, *Tg*Cox2a has homology to Cox2 in other organisms and *Tg*ApiCox25 has homology to the bovine Cox3 module subunit Cox6a [8]. We generated rHA-*Tg*ApiCox13 parasite lines in which we integrated FLAG tags into the 3' ends of the genes encoding *Tg*Cox2a or *Tg*ApiCox25 (S6A–S6D Fig). As a control, we also generated a rHA-*Tg*ApiCox13 parasite line in which we integrated a FLAG tag into the 3' end of the gene encoding the known Complex III protein *Tg*MPPα (S6E and S6F Fig). We cultured rHA-*Tg*ApiCox13/*Tg*ApiCox25-FLAG, rHA-*Tg*ApiCox13/*Tg*Cox2a-FLAG and rHA-*Tg*ApiCox13/*Tg*MPPα-FLAG parasites in the absence of ATc or in the presence of ATc for 1–3 days and assessed complex integrity by BN-PAGE. Knockdown of *Tg*ApiCox13 significantly depleted both the 600 kDa *Tg*ApiCox25-FLAG-containing complex (Fig 5A) and the 600 kDa *Tg*Cox2a-FLAG-containing complex (Fig 5B), but, unlike what we observed upon *Tg*ApiCox25 knockdown [8] (Fig 2C), did not result in the formation of smaller complexes. This implies that subunits of both the putative Cox2 and Cox3 modules of Complex IV are lost upon *Tg*ApiCox13 knockdown. By contrast, the ~675 kDa *Tg*MPPα-FLAG-containing complex (Complex III) was not significantly affected (Fig 5C), suggesting that Complex IV integrity is specifically compromised by *Tg*ApiCox13 knockdown.

We wondered whether the abundance of *Tg*ApiCox25 or *Tg*Cox2a protein monomers decreased upon *Tg*ApiCox13 knockdown. We assessed this by separating proteins extracted from parasites cultured in the absence of ATc or the presence of ATc for 1–3 days by

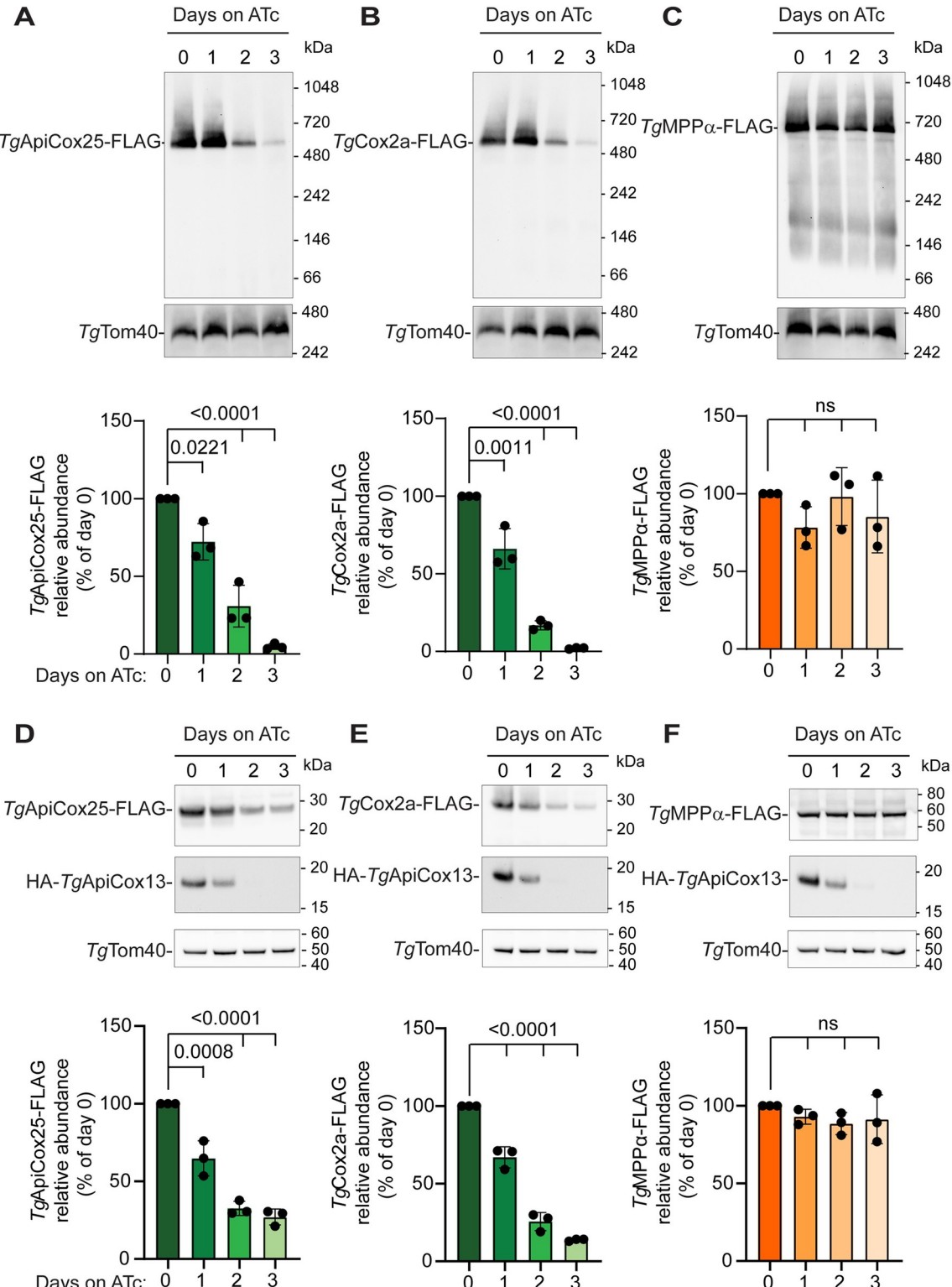

**Fig 5. *Tg*ApiCox13 is important for integrity of Complex IV. (A-C)** Western blots of proteins extracted from **(A)** rHA-*Tg*ApiCox13/*Tg*ApiCox25-FLAG, **(B)** rHA-*Tg*ApiCox13/*Tg*Cox2a-FLAG, and **(C)** rHA-*Tg*ApiCox13/*Tg*MPPα-FLAG parasites cultured in the absence of ATc or in the presence of ATc for 1–3 days. Samples were prepared in 1% (v/v) TX-100, separated by BN-PAGE, and detected with anti-FLAG and anti-*Tg*Tom40 antibodies. **(D-F)** Western blots of proteins extracted from **(D)** rHA-*Tg*ApiCox13/*Tg*ApiCox25-FLAG, **(E)** rHA-*Tg*ApiCox13/*Tg*Cox2a-FLAG, and **(F)** rHA-*Tg*ApiCox13/*Tg*MPPα-FLAG parasites

cultured as described for (A-C). Samples were separated by SDS-PAGE, and probed with anti-HA, anti-FLAG and anti-*Tg*Tom40 antibodies. Western blots are representative of three independent experiments, with matched BN-PAGE and SDS-PAGE samples harvested from the same experiment. The abundance of FLAG-tagged proteins was quantified, normalised relative to the *Tg*Tom40 loading control and expressed as a percentage of the abundance on 0 days ATc. These data are shown as graphs below each blot. Bars represent the mean ± SD of three independent experiments, with each replicate shown as a dot. ANOVA followed by Tukey's multiple comparisons test was performed with relevant *p*-values shown. Colours match those in Fig 4A with Complex IV subunits in green and the Complex III subunit in orange.

SDS-PAGE, and probing for each protein by western blotting. We observed that the abundance of both *Tg*ApiCox25-FLAG and *Tg*Cox2a-FLAG decreased significantly (by ~75%) upon *Tg*ApiCox13 knockdown (Fig 5D and 5E), whereas the abundance of *Tg*MPPα-FLAG (Fig 5F) and the *Tg*Tom40 loading controls remained constant. Together, these data indicate that loss of *Tg*ApiCox13 leads to a loss of Complex IV integrity, and concomitant depletion of Complex IV proteins. This loss of integrity explains the observed defects in Complex IV activity when *Tg*ApiCox13 is depleted.

## *Tg*ApiCox13 possesses a transmembrane domain that is both necessary and sufficient for mitochondrial targeting

Our data indicate that, unlike its human homolog *Hs*MiNT, *Tg*ApiCox13 is a component of Complex IV. Our initial bioinformatics analyses indicated that, while *Hs*MiNT possesses a predicted mitochondrial targeting peptide at its N-terminus, *Tg*ApiCox13 lacks this feature (Fig 1A). Instead, *Tg*ApiCox13 contains a predicted transmembrane domain at its C-terminus, a feature that is not present in the *Hs*MiNT protein (Fig 1A and 1B; S1 Table). We hypothesised that the putative transmembrane domain of *Tg*ApiCox13 may have a role in mitochondrial targeting and/or association with Complex IV.

We first sought to determine whether *Tg*ApiCox13 is an integral membrane protein. We extracted proteins from parasites expressing HA-*Tg*ApiCox13 in alkaline sodium carbonate. We observed that the control peripheral membrane protein *Tg*ATPβ was present solely in the soluble supernatant of the sodium carbonate extraction and the control transmembrane protein *Tg*Tom40 was present solely in the membrane pellet. By comparison, HA-*Tg*ApiCox13 was enriched in the pellet fraction, although some was also found in the supernatant fraction (Fig 6A). These data are consistent with *Tg*ApiCox13 being an integral membrane protein, although the presence of some HA-*Tg*ApiCox13 protein in the soluble fraction may indicate that its putative transmembrane domain is only moderately hydrophobic [29].

To determine whether the transmembrane domain of *Tg*ApiCox13 is necessary for mitochondrial targeting, we introduced a constitutively-expressed monomeric Neon Green (mNG)-tagged copy of *Tg*ApiCox13 (mNG-*Tg*ApiCox13) and a variant of *Tg*ApiCox13 lacking the transmembrane domain (mNG-*Tg*ApiCox13$^{\Delta TM}$) into *T. gondii* parasites. To verify their expression, we performed SDS-PAGE and western blotting, which revealed that mNG-*Tg*ApiCox13 is expressed as a ~40 kDa protein (Fig 6B) and mNG-*Tg*ApiCox13$^{\Delta TM}$ as a ~37 kDa protein (Fig 6C). To assess the cellular localisation of these constructs, we transiently expressed a mitochondrially-targeted red fluorescent protein (mito-RFP) in mNG-*Tg*ApiCox13 and mNG-*Tg*ApiCox13$^{\Delta TM}$ parasites and performed fluorescence microscopy analysis. While mNG-*Tg*ApiCox13 localised to the mitochondrion (Fig 6B), mNG-*Tg*ApiCox13$^{\Delta TM}$ instead localized throughout the parasite cytosol (Fig 6C).

To assess whether the transmembrane domain is sufficient for mitochondrial targeting, we fused the transmembrane domain of *Tg*ApiCox13 to the C-terminus of mNG (mNG-TM). Western blotting detected expression of mNG-TM as a ~28 kDa protein and fluorescence microscopy analysis revealed that mNG-TM localised to the mitochondrion (Fig 6D). Taken

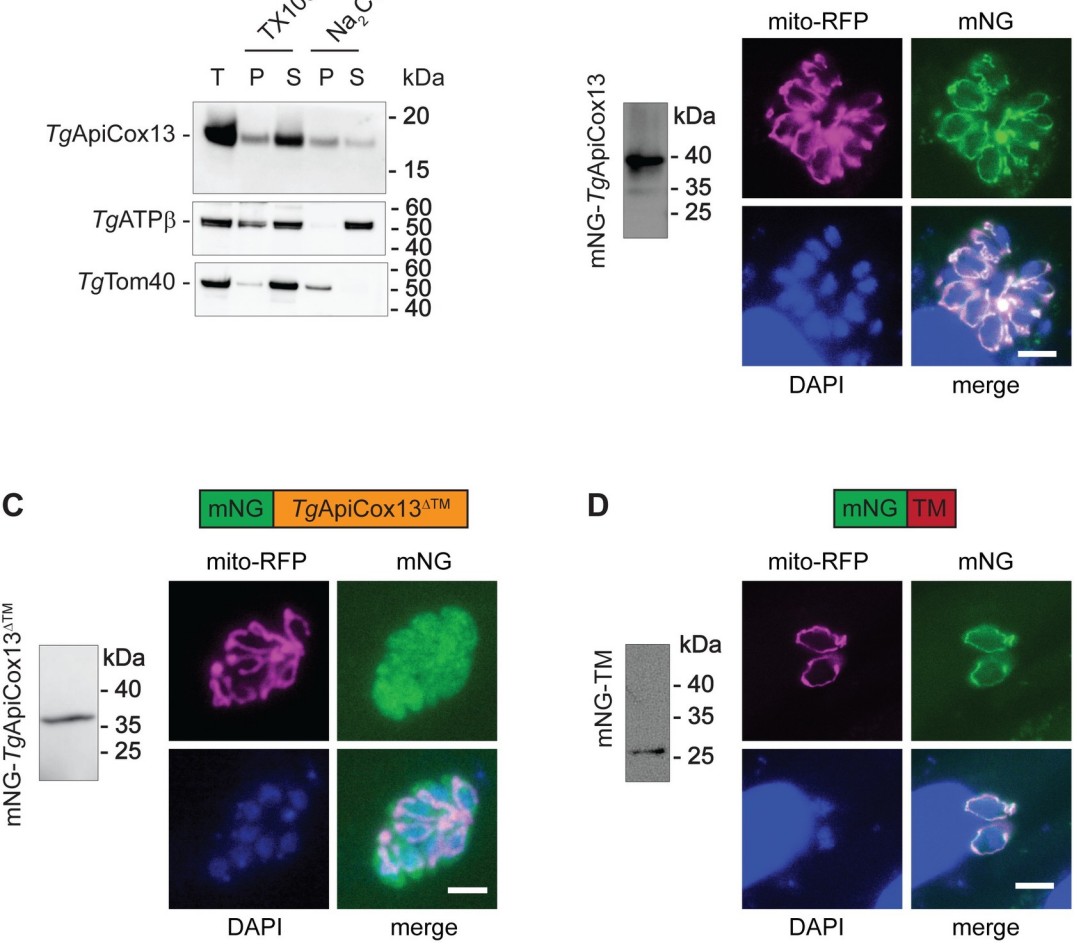

**Fig 6. *Tg*ApiCox13 possesses a transmembrane domain that is both necessary and sufficient for mitochondrial targeting.**
**(A)** Western blot of proteins that were extracted from rHA-*Tg*ApiCox13 parasites and either fractionated into soluble (S) and detergent insoluble phases (P) with TX100 solubilisation, or fractionated into soluble (S) and pellet (P) fractions by alkaline (pH 11) sodium carbonate (Na$_2$CO$_3$) treatment. Total protein extracts (T) are shown in the first lane. Samples were separated by SDS-PAGE, and detected with anti-HA, anti-*Tg*ATPβ or anti-*Tg*Tom40 antibodies. **(B-D)** Western blot and fluorescence microscopy analysis of **(B)** mNG-*Tg*ApiCox13, **(C)** mNG-*Tg*ApiCox13$^{\Delta TM}$, and **(D)** mNG-TM parasites. Proteins were extracted from the parasites, separated by SDS-PAGE and probed with anti-mNG antibodies. Fluorescence microscopy images depict two to eight parasites in a vacuole, detecting monomeric NeonGreen (mNG, green), mitochondrially-targeted red fluorescent protein (mito-RFP, magenta), and the nucleus (DAPI, blue), with overlap shown in the merged images. Scale bars represent 5 μm.

together, these data indicate *Tg*ApiCox13 is an integral membrane protein, with the transmembrane domain being both necessary and sufficient for mitochondrial targeting.

## Fe-S cluster binding by *Tg*ApiCox13 is important for Complex IV function

The two CDGSH motifs of *Hs*MiNT that mediate Fe-S cluster binding are highly conserved in *Tg*ApiCox13 and most other homologs (Fig 1A) [16]. We have shown that recombinant *Tg*ApiCox13 is able to bind Fe-S clusters and that mutating the histidine residues in either or both CDGSH motifs to glutamine abolished Fe-S cluster binding when expressed in *E. coli* (Fig 1C).

This raises questions regarding the role and importance of the Fe-S clusters for *Tg*ApiCox13 function in Complex IV of *T. gondii*.

To begin to address this question, we tested what happens to Complex IV activity when mitochondrial Fe-S cluster biogenesis is disrupted. To do this, we utilised a parasite line generated previously [25], in which the cysteine desulfurase (*Tg*NFS1) that catalyzes the first step of the mitochondrial Fe-S cluster synthesis pathway can be knocked down by the addition of ATc. We have previously shown that the $O_2$ consumption of intact parasites is compromised upon knockdown of *Tg*NFS1 [25], which is unsurprising given the importance of Fe-S cluster biogenesis for catalytic subunits of ETC Complexes II and III. To assess the functionality of ETC Complex IV, we grew WT and r*Tg*NFS1 parasites in the absence of ATc or in the presence of ATc for 1–3 days, starved and permeabilized them, and assessed their responses to malate and TMPD using the Seahorse XFe96 flux analyzer assay (Fig 7A). The first three baseline mOCR readings taken before injection of substrate were close to zero (Fig 7B), indicating that endogenous substrates had been successfully depleted. The injection of malate resulted in an instantaneous increase in OCR in WT parasites cultured in the absence or presence of ATc (blue), and in r*Tg*NFS1 parasites cultured in the absence of ATc (dark purple; Fig 7B; quantified in Fig 7C). By contrast, the malate-elicited mOCR of r*Tg*NFS1 parasites decreased following 2–3 days incubation on ATc (Fig 7B and 7C) when *Tg*NFS1 is depleted [25]. These results indicated that ETC function is decreased upon *Tg*NFS1 knockdown, consistent with our previous observations [25].

To determine whether Complex IV activity was compromised upon *Tg*NFS1 knockdown, we next injected the CytC substrate TMPD (Fig 7A; [10,24]). TMPD elicited $O_2$ consumption in WT parasites cultured in the absence or presence of ATc, and in r*Tg*NFS1 parasites cultured in the absence of ATc or for one day on ATc. By contrast, we observed that r*Tg*NFS1 parasites cultured for 2 or 3 days on ATc had a significantly lower response to TMPD (Fig 7B; quantified in Fig 7D). Calculating the fold stimulation of mOCR by TMPD relative to malate gave values close to 1 (Fig 7E), indicating that TMPD is unable to stimulate mOCR beyond basal levels upon *Tg*NFS1 knockdown. Together these data indicate that loss of *Tg*NFS1 leads to an ETC defect downstream of CytC (*i.e.* in Complex IV). We conclude that mitochondrial Fe-S cluster biogenesis is required for Complex IV activity in *T. gondii*.

Fe-S cluster binding can be important for the stability of Fe-S cluster proteins [25]. We therefore wondered whether *Tg*ApiCox13 protein abundance, and abundance of Complex IV more generally, is reduced upon the depletion of *Tg*NFS1. To test this, we FLAG tagged *Tg*ApiCox13 in the r*Tg*NFS1 line (S7 Fig), grew parasites in the absence of ATc or in the presence of ATc for 1–3 days, and harvested proteins for analysis. We separated proteins by BN-PAGE, and performed western blotting using anti-FLAG antibodies. This revealed that the abundance of the ~600 kDa FLAG-*Tg*ApiCox13-containing complex (*i.e.* Complex IV) is significantly reduced upon *Tg*NFS1 knockdown (Fig 7F). Likewise, SDS-PAGE-based western blotting revealed that the *Tg*ApiCox13 protein itself is also significantly reduced upon *Tg*NFS1 knockdown (Fig 7G). These results are consistent with the hypothesis that Fe-S cluster binding is important for the stability of *Tg*ApiCox13, and is in turn required for Complex IV activity.

To investigate this hypothesis directly, we set out to test the importance of the two Fe-S cluster binding sites of *Tg*ApiCox13 for protein function. We complemented rHA-*Tg*ApiCox13 parasites with versions of the *Tg*ApiCox13 protein in which the histidine residue of either or both of the CDGSH Fe-S cluster binding sites was mutated to glutamine. To enable detection of Complex IV in the resulting parasite line, we undertook these complementations in rHA-*Tg*ApiCox13/*Tg*ApiCox25-GFP parasites, in which the known Complex IV protein *Tg*ApiCox25 had been tagged with GFP (S8 Fig). We constitutively expressed FLAG-tagged copies of the resulting proteins, termed cFLAG-*Tg*ApiCox13[H49Q] (cH49Q), cFLAG-

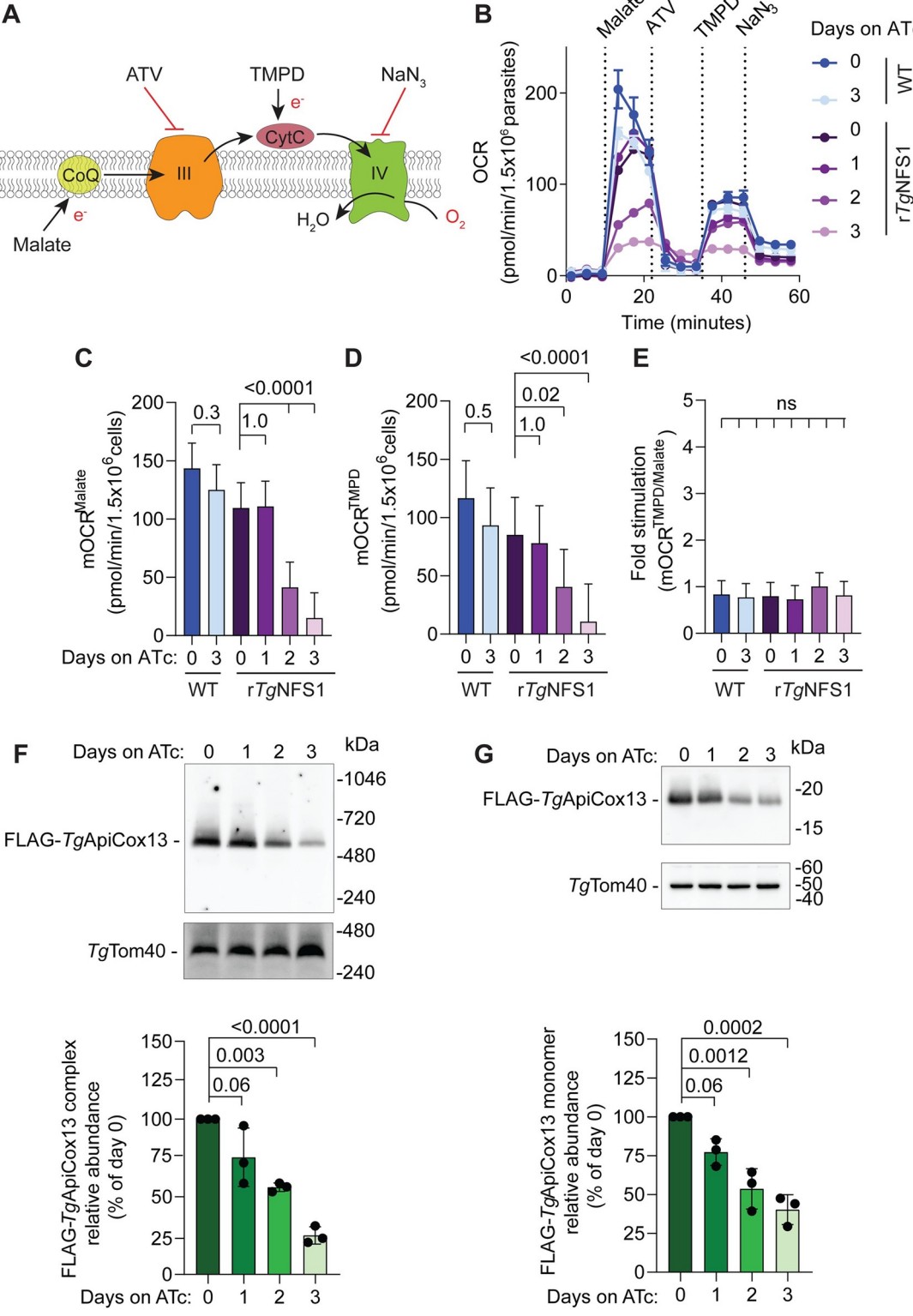

**Fig 7. Mitochondrial Fe-S cluster synthesis is required for Complex IV function and *Tg*ApiCox13 stability. (A)** Schematic of the assay illustrating where substrates and inhibitors act on the ETC. **(B-E)** OCR measurements of WT (blue) and r*Tg*NFS1/ FLAG-*Tg*ApiCox13 (purple) parasites cultured in the absence of ATc or in the presence of ATc for 1–3 days. **(B)** Representative trace depicting OCR over time. Data depict the mean ± SD of 3 technical replicates and are representative of 3 independent experiments. **(C-E)** Quantification of **(C)** malate-elicited mOCR (mOCR$^{Malate}$), **(D)** TMPD-elicited mOCR (mOCR$^{TMPD}$) and

(E) fold stimulation of mOCR elicited by TMPD relative to malate (mOCR$^{TMPD/Malate}$). A linear mixed-effects model was fitted to the data and values depict the estimated marginal mean ± 95% CI of 3 independent experiments. ANOVA followed by Tukey's multiple pairwise comparisons test was performed with relevant *p* values shown. **(F-G)** Western blots of proteins extracted from r*Tg*NFS1/FLAG-*Tg*ApiCox13 parasites cultured in the absence of ATc or in the presence of ATc for 1–3 days. Samples were separated by **(F)** BN-PAGE or **(G)** SDS-PAGE and detected with anti-FLAG and anti-*Tg*Tom40 antibodies. Western blots are representative of three independent experiments, with matched BN-PAGE and SDS-PAGE samples harvested from the same experiment. FLAG-*Tg*ApiCox13 abundance was quantified, normalised relative to *Tg*Tom40 and expressed as a percentage of the abundance on 0 days ATc. These data are shown as graphs below each blot. Bars represent the mean ± SD of three independent experiments, with each replicate shown as a dot. ANOVA followed by Tukey's multiple comparisons test was performed with relevant *p*-values shown.

*Tg*ApiCox13$^{H85Q}$ (cH85Q) and cFLAG-*Tg*ApiCox13$^{H49/85Q}$ (cH49/85Q) in rHA-*Tg*ApiCox13/*Tg*ApiCox25-GFP parasites. We performed western blotting to validate that all three mutated *Tg*ApiCox13 proteins were expressed as ~18 kDa proteins (Fig 8A–8C), and immunofluorescence assays to validate that each localised to the mitochondrion (Fig 8A–8C).

We next investigated the importance of Fe-S cluster binding for *Tg*ApiCox13 function. First, we sought to determine whether *Tg*ApiCox13 Fe-S cluster binding is important for parasite proliferation. We conducted plaque assays in the absence and presence of ATc to assess the ability of the cH49Q, cH85Q and cH49/85Q proteins to complement the proliferation defect observed upon knockdown of the ATc-regulatable HA-*Tg*ApiCox13 protein. We observed that rHA-*Tg*ApiCox13/*Tg*ApiCox25-GFP parasites expressing cH49/85Q exhibited a severe proliferation defect when cultured in the presence of ATc, similar to defect observed in the non-complemented line (Fig 8D). By contrast, parasites expressing cH85Q proliferated normally in the presence of ATc (Fig 8D). Interestingly, an intermediate proliferation defect was observed in rHA-*Tg*ApiCox13/*Tg*ApiCox25-GFP parasites expressing the cH49Q protein and cultured in the presence of ATc (Fig 8D). Together these data indicate that Fe-S cluster binding is critical for the contribution of *Tg*ApiCox13 to parasite proliferation, with the first CDGSH Fe-S cluster binding site of particular importance.

To investigate whether Fe-S cluster binding by *Tg*ApiCox13 is important for Complex IV function, we utilised the Seahorse XFe96 extracellular flux analyzer to measure basal mOCR and thereby assess ETC functionality. We cultured non-complemented rHA-*Tg*ApiCox13/*Tg*ApiCox25-GFP parasites or rHA-*Tg*ApiCox13/*Tg*ApiCox25-GFP parasites complemented with constitutively-expressed FLAG-*Tg*ApiCox13$^{WT}$ (cWT), cH49Q, cH85Q or cH49/85Q proteins in the absence or presence of ATc for 3 days and measured their basal mOCR. As expected, the mOCR of non-complemented rHA-*Tg*ApiCox13/*Tg*ApiCox25-GFP parasite cultured in the presence of ATc was substantially depleted (Fig 8E), whereas mOCR of cWT expressing parasites cultured in ATc was normal. Notably, constitutive expression of the cH49/85Q protein was unable to restore mOCR in parasites cultured in the presence of ATc (Fig 8E). Expression of the cH85Q protein was able to fully complement mOCR upon *Tg*ApiCox13 knockdown, whereas expression of cH49Q resulted in significantly (~50%) lower basal mOCR in the presence of ATc (Fig 8E). Together, these data imply that Fe-S cluster binding is critical for the function of *Tg*ApiCox13 in the ETC, with the first CDGSH Fe-S cluster binding site of particular importance.

We wondered whether Fe-S cluster binding by *Tg*ApiCox13 is important for the association of the *Tg*ApiCox13 protein with Complex IV and for Complex IV integrity. We first examined the abundances of Complex IV proteins in rHA-*Tg*ApiCox13/*Tg*ApiCox25-GFP parasites complemented with constitutively expressed cWT, cH49Q, cH85Q or cH49/85Q proteins. We cultured parasite lines in the absence or presence of ATc for 3 days and harvested proteins for analysis. We compared the abundances of the cH49Q, cH85Q and cH49/85Q proteins to that of cWT by SDS-PAGE and western blotting using anti-FLAG antibodies (Fig 9A). When

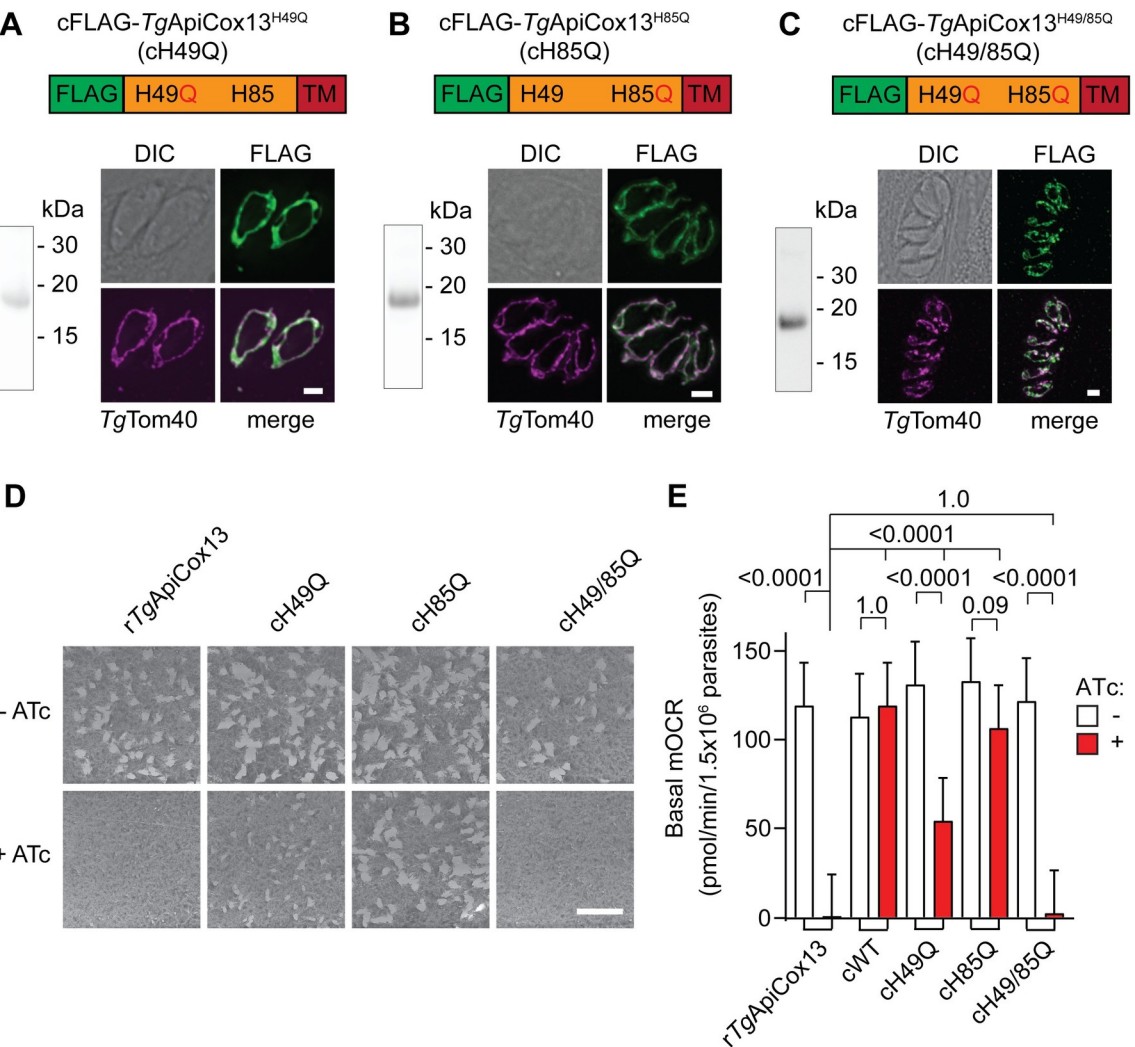

**Fig 8. Fe-S cluster binding by *Tg*ApiCox13 is required for parasite proliferation and ETC activity. (A-C)** Western blots and immunofluorescence assays of rHA-*Tg*ApiCox13/*Tg*ApiCox25-GFP parasites constitutively expressing **(A)** cFLAG-*Tg*ApiCox13^H49Q, (cH49Q) **(B)** cFLAG-*Tg*ApiCox13^H85Q (cH85Q), or **(C)** cFLAG-*Tg*ApiCox13^H49/85Q (cH49/85Q). Proteins were extracted from the parasites, separated by SDS-PAGE, and probed by western blotting with anti-FLAG antibodies (left). Immunofluorescence assays (right) depict two to eight parasites in a vacuole, with FLAG-*Tg*ApiCox13 proteins detected with anti-FLAG (green) and the mitochondrion with anti-*Tg*Tom40 (magenta) antibodies. DIC, differential interference contrast; scale bars represent 2 μm. **(D)** Plaque assays measuring the proliferation of *Tg*ApiCox13/*Tg*ApiCox25-GFP parasites (r*Tg*ApiCox13), or *Tg*ApiCox13/*Tg*ApiCox25-GFP parasites constitutively expressing the cH49Q, cH85Q, or cH49/85Q proteins, and cultured in the absence (top) or presence (bottom) of ATc for 8 days. Assays are from a single experiment and are representative of 3 independent experiments; scale bar represents 1 cm. **(E)** Basal mitochondrial oxygen consumption rate (mOCR) of *Tg*ApiCox13/*Tg*ApiCox25-GFP parasites (r*Tg*ApiCox13), or *Tg*ApiCox13/*Tg*ApiCox25-GFP parasites constitutively expressing the cWT, cH49Q, cH85Q or cH49/85Q proteins, and cultured in the absence (white) or presence (red) of ATc for 3 days. A linear mixed-effects model was fitted to the data and values depict the estimated marginal mean ± 95% CI of three independent experiments. ANOVA followed by Tukey's multiple pairwise comparisons test was performed with relevant *p* values shown.

quantified relative to the *Tg*Tom40 loading control, the abundances of the cH49Q, cH85Q and cH49/85Q proteins were decreased by ~55%, ~15% and ~35%, respectively, compared to the cWT protein in the absence of ATc (Fig 9B, top). However, in the presence of ATc, the abundance of the cH49Q and cH85Q proteins increased to ~75% and ~120% that of the cWT protein, whereas the abundance of the cH49/85Q protein was unchanged (Fig 9B, top). We also found that the abundance of the *Tg*ApiCox25-GFP protein decreased significantly upon ATc-

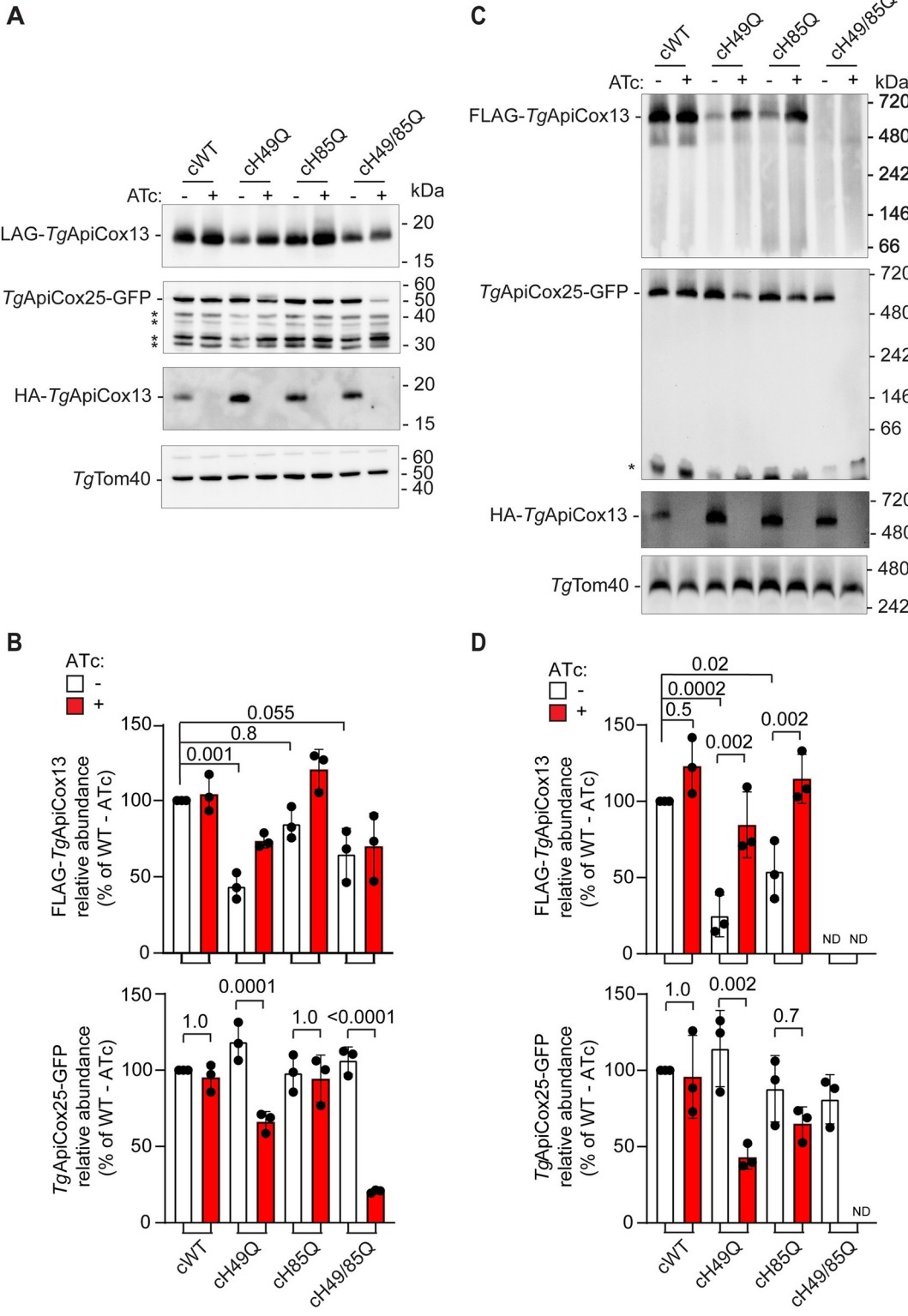

**Fig 9. Fe-S cluster binding by *Tg*ApiCox13 is required for Complex IV stability. (A and C)** Western blot of proteins extracted from rHA-*Tg*ApiCox13/*Tg*ApiCox25-GFP parasites constitutively expressing cFLAG-*Tg*ApiCox13$^{WT}$ (cWT), cFLAG-*Tg*ApiCox13$^{H49Q}$ (cH49Q), cFLAG-*Tg*ApiCox13$^{H85Q}$ (cH85Q) or cFLAG-*Tg*ApiCox13$^{H49/85Q}$ (cH49/85Q) proteins, cultured in the absence (-, white) or presence (+, red) of ATc for 3 days, separated by **(A)** SDS-PAGE or **(C)** BN-PAGE, and probed using anti-FLAG (top), anti-GFP (upper middle), anti-HA (lower middle) or anti-*Tg*Tom40 (bottom) antibodies.

Western blots are representative of 3 independent experiments, with matched BN-PAGE and SDS-PAGE samples harvested from the same experiment. Asterisks in the anti-GFP western blots mark the presence of likely degradation products of the *Tg*ApiCox25-GFP protein. **(B and D)** The abundance of FLAG- (top) and GFP- (bottom) tagged protein **(B)** monomers or **(D)** complexes was quantified, normalised relative to the *Tg*Tom40 loading control and expressed as a percentage of the cWT abundance in the absence of ATc. These quantifications are shown as column graphs with bars representing the mean ± SD of three independent experiments. ANOVA followed by Tukey's multiple comparisons test was performed with relevant *p*-values shown.

mediated HA-*Tg*ApiCox13 knockdown in parasites constitutively expressing the cH49Q (~40% reduced) and cH49/85Q (~75% reduced) proteins (Fig 9B, bottom). By contrast, the abundance of *Tg*ApiCox25 was maintained upon HA-*Tg*ApiCox13 knockdown in parasites constitutively expressing the cH85Q and cWT proteins (Fig 9B, bottom).

We next sought to determine whether Fe-S cluster binding by *Tg*ApiCox13 is important for association of the *Tg*ApiCox13 protein with Complex IV. We cultured rHA-*Tg*ApiCox13/*Tg*A-piCox25-GFP parasites complemented with constitutively expressed cWT, cH49Q, cH85Q or cH49/85Q proteins in the absence or presence of ATc for three days, separated extracted proteins by BN-PAGE, and probed samples by western blotting (Fig 9C). We found that the cWT, cH49Q and cH85Q proteins were part of ~600 kDa protein complexes (Fig 9C), the predicted mass of Complex IV in *T. gondii*. By contrast, the cH49/85Q protein was not present in a detectable protein complex. When quantified, the abundances of the cH49Q- and cH85Q-containing complexes were significantly reduced by ~75% and ~45%, respectively, compared to the cWT-containing complex when parasites were cultured in the absence of ATc (Fig 9D, top). Interestingly, the abundances of both the cH49Q- and cH85Q-containing complexes increased significantly when parasites were cultured in the presence of ATc, to levels close to the abundance of the cWT-containing complex (Fig 9D, top). These data suggest that presence of at least one bound Fe-S cluster is required for *Tg*ApiCox13 to associate with Complex IV. Additionally, the increased abundances of the cH49Q and cH85Q proteins, but not the cWT protein, in Complex IV when ATc-regulated HA-*Tg*ApiCox13 is knocked down suggests the existence of a quality control mechanism by which integration of *Tg*ApiCox13 into Complex IV is dependent on the Fe-S binding status (and perhaps proper or efficient folding) of the protein.

To assess whether Fe-S cluster binding by *Tg*ApiCox13 is important for the integrity of Complex IV, we performed western blotting using anti-GFP antibodies to detect *Tg*ApiCox25 (Fig 9C). While cWT *Tg*ApiCox13 was able to maintain Complex IV integrity to similar levels upon knockdown of native HA-*Tg*ApiCox13, no Complex IV was detectable in the cH49/85Q mutant in the presence of ATc (Fig 9C). When quantified relative to *Tg*Tom40, Complex IV abundance was significantly reduced by ~60% in the cH49Q-expressing parasites upon knockdown of native HA-*Tg*ApiCox13 compared to the absence of ATc (Fig 9D), while the cH85Q mutant exhibited a smaller (~20%) decrease in Complex IV abundance (Fig 9D). These results indicate that binding to at least one Fe-S cluster is important for *Tg*ApiCox13 to associate with Complex IV and, mirroring the effects on parasite proliferation and ETC activity, that the first CDGSH domain is particularly important for the integrity of Complex IV.

We conclude that Fe-S cluster binding by *Tg*ApiCox13 is important for its function, and that the first CDGSH motif is likely of particular importance given that the cH49Q mutant exhibited more severe phenotypes than the cH85Q mutant in parasite proliferation, ETC functionality and Complex IV integrity.

## Does *Tg*ApiCox13 have a function beyond Complex IV?

Human NEET proteins transfer Fe-S clusters to other proteins [19]. Our data thus far have established that *Tg*ApiCox13 has a function in ETC Complex IV, but it is conceivable that

*Tg*ApiCox13 has additional mitochondrial functions beyond Complex IV. We therefore wondered whether Fe-S cluster transfer is an important function of *Tg*ApiCox13.

If *Tg*ApiCox13 can transfer Fe-S clusters, we hypothesised that it could possibly donate them to other Fe-S cluster-containing ETC complex subunits (*e.g.* the Complex II SdhB subunit) or to another mitochondrial Fe-S protein (*e.g.* ferredoxin). We have previously demonstrated that loss of Fe-S cluster synthesis can lead to the depletion of Fe-S proteins [25]. To test whether the loss of *Tg*ApiCox13 leads to depletion of Fe-S proteins, we added a Ty1 epitope tag to the Complex II Fe-S cluster-containing subunit *Tg*SdhB and a FLAG epitope tag to the mitochondrial Fe-S protein ferredoxin (*Tg*mtFDX) in the rHA-*Tg*ApiCox13 parasite line (S9 Fig). As a control, we also epitope tagged these proteins in the r*Tg*NFS1-HA line which is defective in mitochondrial Fe-S cluster assembly (S9 Fig; [25]). We grew these parasites in the absence or presence of ATc for 3 days, extracted samples and separated proteins by SDS-PAGE. Western blotting and subsequent quantification revealed that the abundance of both *Tg*SdhB-Ty1 and *Tg*mtFDX-FLAG was significantly decreased upon knockdown of *Tg*NFS1 (Fig 10A–10C), consistent with these proteins requiring Fe-S clusters for their stability. By contrast, *Tg*mtFDX-FLAG abundance was slightly (~30%) but significantly decreased upon knockdown of *Tg*ApiCox13 (Fig 10C) whereas *Tg*SdhB abundance did not significantly change upon *Tg*ApiCox13 knockdown (Fig 10B). Our data suggest that *Tg*ApiCox13 is not required for the biogenesis of the mitochondrial Fe-S proteins that we examined, although *Tg*ApiCox13 may contribute to optimal *Tg*mtFDX biogenesis or turnover.

We finally sought to investigate the functional consequences of limiting *Tg*ApiCox13's ability to transfer Fe-S clusters. Previous studies of the related Fe-S cluster protein *Hs*NAF-1 have shown that mutating the histidine of the Fe-S cluster binding CDGSH motifs to cysteine impairs Fe-S cluster transfer [30], as the Fe-S cluster is coordinated more tightly by four cysteine residues. We complemented rHA-*Tg*ApiCox13/*Tg*ApiCox25-GFP parasites with a constitutively expressed, FLAG-tagged version of the *Tg*ApiCox13 protein with equivalent mutations and termed the resultant parasite line cFLAG-*Tg*ApiCox13$^{H49/85C}$ (cH49/85C). We performed western blotting to validate cH49/85C expression and an immunofluorescence assay to validate that it localised to the mitochondrion (Fig 11A). We next conducted plaque assays in the absence and presence of ATc to assess the ability of the cH49/85C protein to complement the proliferation defect observed upon knockdown of the ATc-regulatable HA-*Tg*ApiCox13 protein (Fig 11B). We observed that cH49/85C parasites proliferated just as well as parasites complemented with WT *Tg*ApiCox13 (cWT) (Fig 11B). We then sought to investigate whether *Tg*ApiCox13 Fe-S cluster transfer is important for ETC function. We cultured non-complemented rHA-*Tg*ApiCox13/*Tg*ApiCox25-GFP parasites or parasites complemented with cWT or cH49/85C *Tg*ApiCox13 protein in the absence or presence of ATc for 3 days and measured their basal mOCR using the Seahorse XFe96 flux analyser. We observed that cH49/85C expressing parasites were able to fully complement the defect in OCR observed upon loss of HA-*Tg*ApiCox13 (Fig 11C). Although we have no direct evidence that the cH49/85C protein is unable to transfer Fe-S clusters, our data imply that if *Tg*ApiCox13 can transfer Fe-S clusters, this ability does not appear to be critical for its contribution to parasite proliferation or ETC activity.

## Discussion

Fe-S cluster containing proteins have many important roles in mitochondrial biology [31]. Complexes I, II and III of the mitochondrial ETC contain Fe-S clusters that play key roles in electron transfer reactions. By comparison, Complex IV of most eukaryotes lacks Fe-S clusters and instead utilizes heme and copper co-factors for electron transport. In this work, we

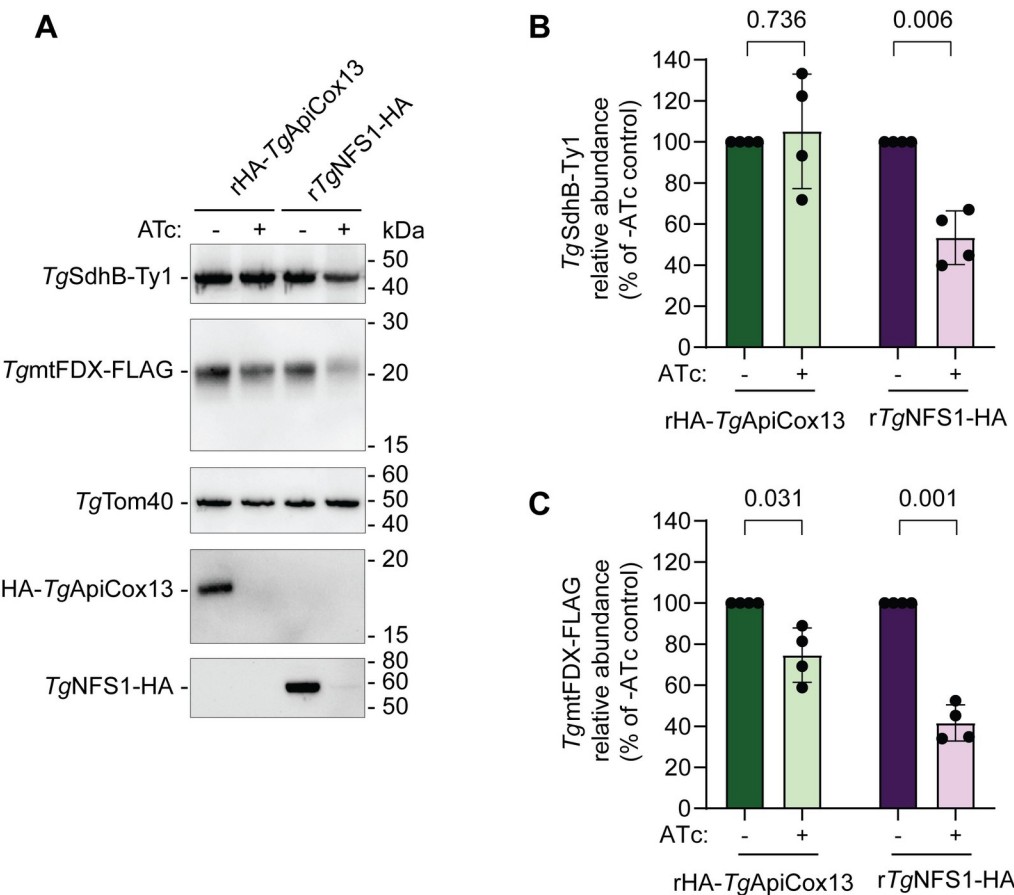

**Fig 10. *Tg*ApiCox13 contributes minimally to the biogenesis of mitochondrial Fe-S proteins. (A)** Western blot of proteins extracted from rHA-*Tg*ApiCox13/*Tg*SdhB-Ty1/*Tg*mtFDX-FLAG or r*Tg*NFS1-HA/*Tg*SdhB-Ty1/*Tg*mtFDX-FLAG parasites cultured in the absence (-) or presence (+) of ATc for 3 days and separated by SDS-PAGE. Western blots were probed using anti-Ty1 (top), anti-FLAG (upper middle), anti-*Tg*Tom40 (middle), or anti-HA (lower middle and bottom) antibodies. Western blots are representative of four independent experiments. **(B and C)** The abundances of the Fe-S cluster proteins *Tg*SdhB-Ty1 (B) and *Tg*mtFDX-FLAG (C) were quantified from the western blots described in (A), normalised relative to the *Tg*Tom40 loading control and expressed as a percentage of the—ATc control. These quantifications are shown as column graphs with bars representing the mean ± SD of four independent experiments. Unpaired t-tests with Welch's correction were performed with relevant *p*-values shown.

demonstrate that the Fe-S cluster-containing protein *Tg*ApiCox13 contributes to Complex IV function and integrity and is important for the proliferation of *T. gondii* parasites. To our knowledge, this represents the first functional characterization of an Fe-S cluster protein sub-unit in Complex IV.

*Tg*ApiCox13 is homologous to the human MiNT protein, which is an Fe-S cluster binding protein containing two CDGSH motifs that enable the coordination of two Fe-S clusters. Nota-bly, *Hs*MiNT is not part of Complex IV in humans and instead functions in mitochondrial iron homeostasis [16]. The two CDGSH motifs of *Hs*MiNT are conserved in *Tg*ApiCox13, and we demonstrate that they contribute to *Tg*ApiCox13 protein function in Complex IV of *T. gon-dii*. Curiously, our data imply that the first CDGSH motif is more important than the second CDGSH motif for *Tg*ApiCox13 function (Figs 8 and 9). The first motif is conserved through-out the ApiCox13/MiNT family, whereas the second motif lacks key cysteine and/or histidine residues required for Fe-S cluster binding in the ApiCox13 homologues of *P. falciparum* and the dinoflagellate *Symbiodinium* (Fig 1A). *Pf*ApiCox13 was, nonetheless, detected as a

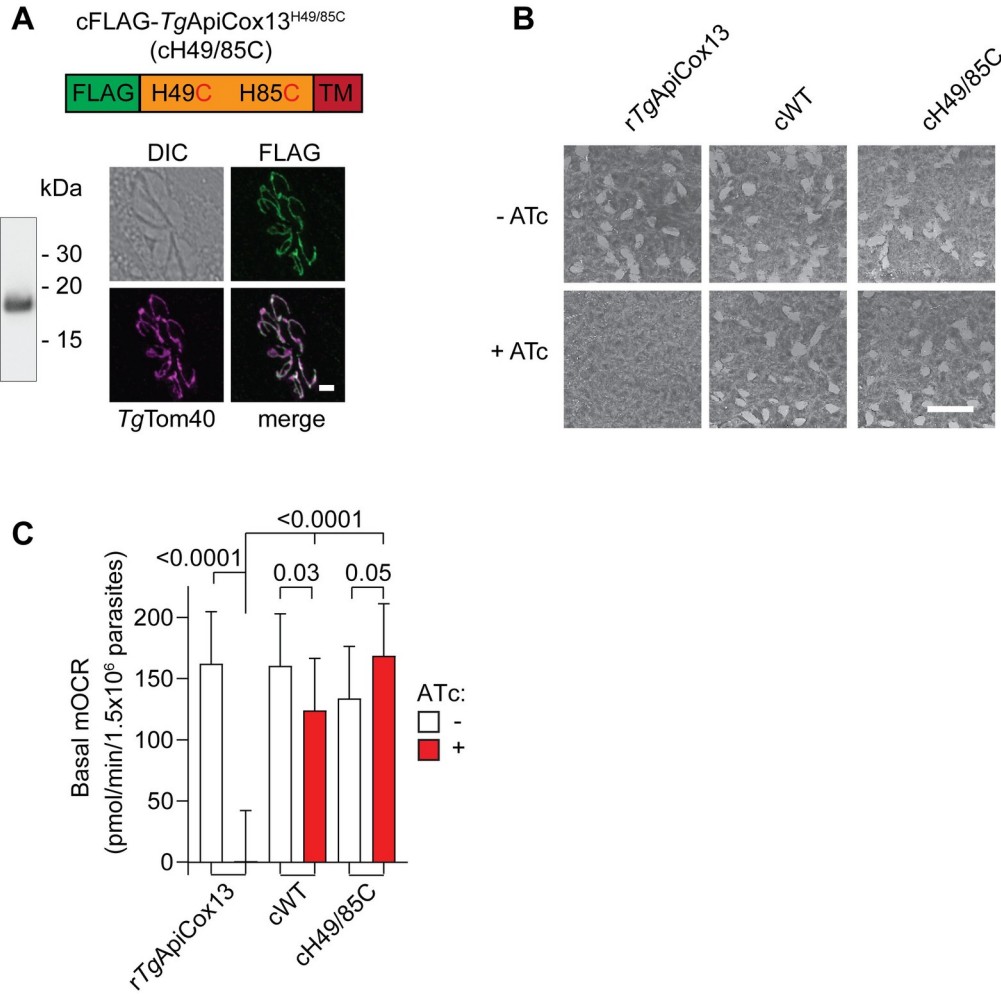

**Fig 11. Fe-S cluster transfer ability is unlikely to be essential for *Tg*ApiCox13 function. (A)** Western blot and immunofluorescence assay of rHA-*Tg*ApiCox13/*Tg*ApiCox25-GFP parasites constitutively expressing cFLAG-*Tg*ApiCox13^H49/85C (cH49/85C). Proteins were extracted from the parasites, separated by SDS-PAGE, and probed by western blotting with anti-FLAG antibodies (left). Immunofluorescence assay (right) depicts eight parasites in a vacuole, with FLAG- cH49/85C detected with anti-FLAG (green) and the mitochondrion with anti-*Tg*Tom40 (magenta) antibodies. DIC, differential interference contrast; scale bar represents 2 μm. **(B)** Plaque assays measuring the proliferation of *Tg*ApiCox13/*Tg*ApiCox25-GFP parasites (r*Tg*ApiCox13), or *Tg*ApiCox13/*Tg*ApiCox25-GFP parasites constitutively expressing WT *Tg*ApiCox13 (cWT) or cH49/85C proteins, and cultured in the absence (top) or presence (bottom) of ATc for 8 days. Assays are from a single experiment and are representative of 3 independent experiments; scale bar represents 1 cm. **(C)** Basal mitochondrial oxygen consumption rate (mOCR) of *Tg*ApiCox13/*Tg*ApiCox25-GFP parasites (r*Tg*ApiCox13), or *Tg*ApiCox13/*Tg*ApiCox25-GFP parasites constitutively expressing the cWT or cH49/85Q proteins, and cultured in the absence (white) or presence (red) of ATc for 3 days. A linear mixed-effects model was fitted to the data and values depict the estimated marginal mean ± 95% CI of three independent experiments. ANOVA followed by Tukey's multiple pairwise comparisons test was performed with relevant *p* values shown.

component of Complex IV in a recent proteomic analysis of the *P. falciparum* ETC complexes [12]. Together, these data imply that the second Fe-S binding site of ApiCox13 proteins is not critical for Complex IV function.

Our data indicate that the Fe-S clusters are important for the function of *Tg*ApiCox13 and, in turn, the stability of Complex IV (Figs 8 and 9). We also observed that the abundance of both the *Tg*ApiCox13 monomer and complex was decreased when the Fe-S cluster assembly

protein *Tg*NFS1 was knocked down (Fig 7F and 7G). This is consistent with the findings of a recent study [23], which reported decreased abundance of *Tg*ApiCox13 and 80% of other identified Complex IV subunits in *T. gondii* upon knockdown of the mitochondrial Fe-S cluster scaffold protein *Tg*ISU1. Together, these findings support the hypothesis that mitochondrial Fe-S cluster synthesis contributes to Complex IV stability and abundance through modulating Fe-S cluster integration into *Tg*ApiCox13.

Given that loss of its Fe-S clusters led to decreased Complex IV stability and activity, we speculate that *Tg*ApiCox13 has a role in the iron-dependent regulation of ETC function. Iron is an important micronutrient for many intracellular parasites, including *T. gondii* [32], and modulation of intracellular iron levels is critical for the survival of *T. gondii* parasites [33]. Directly linking iron metabolism in the parasite and ETC activity through the Fe-S clusters of *Tg*ApiCox13 provides these parasites with a means of regulating the ETC (and associated processes such as ATP synthesis and pyrimidine synthesis) in response to iron availability. For example, in iron-limiting conditions, we anticipate that parasites will limit ETC activity. ETC inhibition and inhibition of mitochondrial Fe-S synthesis has been linked to parasites entering a slow-growing, so-called bradyzoite stage of the parasite life cycle [10, 23, 34], which is critical for latent infection of hosts. Whether iron limitation results in bradyzoite differentiation will be an interesting area for future studies.

The mitochondrial ETC is a major producer of reactive oxygen species (ROS; [35]). Control of ETC activity is therefore critical for modulating mitochondrial ROS production in cells. Fe-S clusters in mitochondrial proteins such as aconitase and ferrochelatase have been shown to exhibit increased lability upon exposure to ROS [36], and are lost from the protein. It is conceivable that reaction of ROS with the Fe-S clusters of *Tg*ApiCox13 could likewise lead to loss of Fe-S clusters and destabilisation of the protein and Complex IV. In this manner, the Fe-S clusters of *Tg*ApiCox13 could enable Complex IV activity to be decreased (and therefore overall ETC activity to be depleted) in response to high mitochondrial ROS levels. In turn, this would limit ETC-generated ROS, thus modulating mitochondrial ROS production in the parasite. Whether ROS causes lability of the Fe-S clusters in *Tg*ApiCox13, and whether this leads to depletion of Complex IV integrity and ETC function, are questions for the future.

A common function of NEET proteins is to transfer Fe-S clusters to other proteins; for instance, *Hs*MiNT can transfer Fe-S clusters to mitochondrial ferredoxin [16]. We observed that the abundance of the mitochondrial Fe-S protein *Tg*SdhB was unchanged upon knockdown of *Tg*ApiCox13, and that another mitochondrial Fe-S protein, *Tg*mtFDX, decreased by ~30% (Fig 10). These observations are in contrast to knockdown of mitochondrial Fe-S cluster synthesis, which led to a much greater depletion in abundance of both *Tg*SdhB and *Tg*mtFDX (Fig 10). Our data, therefore, indicate that *Tg*ApiCox13 does not contribute to bulk biogenesis of mitochondrial Fe-S proteins, although may have a smaller role in mediating *Tg*mtFDX biogenesis or stability. We also found that a *Tg*ApiCox13 variant with H49/85C mutations–which is predicted to stabilize Fe-S cluster binding and prevent Fe-S cluster transfer [30]–was able to fully complement the loss of *Tg*ApiCox13 (Fig 11). Taken together, our data suggest that if *Tg*ApiCox13 does transfer Fe-S clusters, this ability is unlikely to be an essential function in the parasite. Exploring the ability of *Tg*ApiCox13 to transfer Fe-S clusters in more depth is an avenue for future research.

A possible role for human NEET proteins in transferring Fe-S clusters from the mitochondrion to cytosol has been proposed. Both NAF-1 and the mitochondrial outer membrane protein mitoNEET can transfer their Fe-S clusters to the cytosolic iron-sulfur cluster assembly (CIA) pathway protein anamorsin (also known as Dre2 or CIAPIN1) [37], and mitoNEET can transfer its Fe-S clusters to NAF-1 [38]. A recent study proposed that human MiNT, the homolog of *Tg*ApiCox13, can directly transfer Fe-S clusters to mitoNEET via interaction of these

proteins at opposite ends of the transmembrane channel of the OMM voltage-dependent anion channel 1 (VDAC1) [39]. However, how matrix-localised MiNT gains access to these OMM proteins through the inner mitochondrial membrane is unclear. It is conceivable that *Tg*ApiCox13 has a role in transferring mitochondrially-derived Fe-S clusters to the cytosolic Fe-S assembly pathway, which in *T. gondii* localizes to the cytosolic face of the outer mitochondrial membrane [25]. Curiously, we found that knockdown of the Complex IV subunits *Tg*ApiCox13 and *Tg*ApiCox25 was less reversible than knockdown of the Complex III protein *Tg*QCR11 (S5 Fig). One possible explanation for this is that Complex IV has additional functions beyond the ETC. Further investigating the possible roles of *Tg*ApiCox13 and Complex IV in contributing to cytosolic Fe-S protein biogenesis will therefore be of particular interest.

Fe-S clusters in Complexes I, II and III of the ETC are essential for the electron transfer reactions that occur in these complexes [3–5,40]. Could the Fe-S clusters of *Tg*ApiCox13 also participate in the electron transfer reaction through Complex IV to oxygen? Notably, recent structural analyses of Complex IV in the ciliate *Tetrahymena* identified an ApiCox13 homolog (COXFS) in this complex [13, 14]. COXFS is localised on the mitochondrial matrix side and anchored into Complex IV via a C-terminal transmembrane helix (S10A Fig) [13]. This transmembrane helix makes contact with several other helices that are packed close to the two transmembrane helices of the catalytic subunit COX2. The matrix-localised portion of the COXFS protein (containing the two Fe-S clusters) makes contact with many non-catalytic subunits within Complex IV but is not in direct contact with the catalytic subunits COX1 and COX2 (S10B Fig). We calculated the distance between the known electron transferring functional groups of Complex IV of *Tetrahymena*, and found these to vary between 4.8 to 19.5 Å (S10C Fig). By contrast, the distance between the Fe-S clusters of COXFS and the other redox sites is much larger (between 50.7 and 54.8 Å; S10C Fig). This distance is much larger than the distance for most electron transfer reactions in protein complexes [41], arguing against COXFS/ApiCox13 having a direct role in electron transport through Complex IV. Notably, the COXFS Fe-S clusters appear to be accessible to the matrix solvent courtesy of a cleft in Complex IV (S10A Fig), which could allow COXFS to respond to a mitochondrial signal (*e.g.* ROS or changes in iron abundance). Resolving the structure of Complex IV from an apicomplexan parasite would provide further structural insights into these possibilities.

Taken together, our study highlights the key role of an Fe-S protein in modulating the function of Complex IV in *T. gondii*. The existence of ApiCox13 homologs in other apicomplexans and their closest free-living relatives (chromerids, dinoflagellates and ciliates), highlights how key mitochondrial processes have undergone considerable divergence in a major eukaryotic lineage.

## Methods

### Cloning, expression and purification of recombinant *Tg*ApiCox13

To generate WT and point mutations of recombinantly-expressed, His-tagged *Tg*ApiCox13, the open reading frame sequence encoding amino acids 1–93 of *Tg*ApiCox13 (TGGT1_254030) was amplified by reverse transcription (RT)-PCR using the primers *Tg*ApiCox13$^{1–93}$ ORF fwd and rvs (S2 Table) and cloned into the *Nde*I and *Hin*dIII sites of the pCold-SUMOa expression vector. The point mutations in *Tg*ApiCox13$^{1-93}$ were introduced by site-directed mutagenesis using Fast Mutagenesis Kit V2 (Vazyme #C214-02, Nanjing, China; primers with the respective nucleotide exchanges are shown in S2 Table). All constructs were sequenced for accuracy prior to further use, using the primers *Tg*ApiCox13$^{1–93}$ seq fwd and rvs (S2 Table). All resulting plasmids were transformed into *Escherichia coli* BL21(DE3) cells. Protein expression was induced by the addition of 1 mM IPTG at 18°C for 24 h. The induced

cells were collected and resuspended into 30 ml of cold buffer A (20 mM Tris-HCl, pH 8.0, 0.5 M NaCl) and then disrupted using a low temperature ultra-high pressure continuous flow cell disrupter (JN-3000 PLUS, JNBIO). Cell debris was removed by centrifuging at 20,000 ×$g$ for 45 min, and the supernatants were applied to 2 ml of Superflow nickel-nitrilotriacetic acid aga-rose affinity column (Qiagen) attached to a ÄKTA-fast protein liquid chromatography system (ÄKTA Purifier 100, GE, CT). The column was washed with 3 column volumes of buffer A and 3 column volumes of buffer B (20 mM Tris-HCl, pH 8.0, 500 mM NaCl, 20 mM imidaz-ole). The recombinant proteins were eluted with buffer C (20 mM Tris-HCl, pH 8.0, 500 mM NaCl, 500 mM imidazole) and then added to a Hitrap-desalting column (5.0 ml) and equili-brated with buffer A to remove imidazole. The purity of all purified proteins was assessed using SDS-PAGE followed by Coomassie blue staining, and the protein concentration was determined using an extinction coefficient at 280 nm of 47.19 mM$^{-1}$ cm$^{-1}$. The presence of iron-sulfur clusters in the *Tg*ApiCox13 protein variants were evaluated using U-3900 UV-visi-ble spectrophotometer (HITACHI, Japan).

## Iron content analyses

Iron content of recombinant proteins were measured as described previously [42] with minor modification. Briefly, to detect the iron content, purified proteins were incubated with 1 mM ferroZine and 4 mM L-cysteine at 85°C for 30 min followed by centrifugation at 13,000 ×$g$ for 5 min to harvest the supernatants. The concentration of the iron-ferroZine complex was mea-sured at 564 nm using an extinction coefficient of 27.9 mM$^{-1}$ cm$^{-1}$. The *E. coli* endonuclease III (Nth) was used as a standard. Statistical differences were tested using ANOVA followed by Tukey's multiple comparisons test in GraphPad Prism.

## Parasite culture

*T. gondii* parasites were cultured in human foreskin fibroblast (HFF) cells in modified Dulbec-co's Modified Eagle's Medium (DMEM) containing 2 g/L sodium bicarbonate, supplemented with 1% (v/v) fetal calf serum, 0.2 mM glutamine, 50 U/mL penicillin, 50 μg/mL streptomycin, 10 μg/mL gentamicin and 0.25 μg/mL amphotericin b. Parasite cultures were incubated in a humidified 37°C incubator at 5% $CO_2$. Where appropriate, 0.5 μg/mL of anhydrotetracycline (ATc) or an equivalent volume of ethanol (vehicle control) was added.

## Generation of modified parasites

Using CRIPSR/Cas9-based genome editing we simultaneously introduced an HA tag into the 5' end of the *Tg*ApiCox13 open reading frame and replaced its native promoter with an anhy-drotetracycline (ATc)-regulatable promoter. We introduced a single guide RNA targeting the 5' end of the open reading frame of *Tg*ApiCox13 into the pSAG1::Cas9-U6-UPRT vector (Addgene plasmid 54467; [43]) using Q5-site directed mutagenesis according to the manufac-turer's instructions (New England Biolabs). We performed the Q5 reaction using *Tg*ApiCox13 5' CRISPR fwd primer and a generic CRISPR rvs primer (S2 Table), confirming successful plasmid modification by Sanger sequencing. We amplified a donor DNA sequence encoding an HA tag and a ATc-regulatable promoter together with 50 bp flanks of *Tg*ApiCox13 using the pPR2-HA$_3$ vector [44] as a template along with *Tg*ApiCox13 promotor replacement (pro rep) fwd and rvs primers (S2 Table).

TATiΔ*ku80* line parasites [45] were transfected with the combined plasmid and donor DNA by electroporation using a single 1.5 kV pulse at 25 μF capacitance and 50 Ω resistance [46]. After two days, green fluorescent protein (GFP) positive parasites, expressing the Cas9-GFP encoded on the vector, were sorted into wells of a 96-well plate using a

FACSMelody cell sorter (BD Biosciences). We identified wells with single plaques, indicating clonal parasite populations, and screened these by polymerase chain reaction (PCR) to determine which clones had integrated the epitope tag and regulatable promoter, screening with *Tg*ApiCox13 screen fwd and rvs primers (S2 Table). We termed the resulting parasite line 'rHA-*Tg*ApiCox13'.

To introduce a FLAG tag into the 5' open reading frame of *Tg*ApiCox13, we used the *Tg*ApiCox13 CRISPR vector described above. We amplified donor DNA encoding a FLAG epitope tag and 50bp flanks of *Tg*ApiCox13 using the N-terminal FLAG gBlock and *Tg*ApiCox13 FLAG fwd and rvs primers (S2 Table). We combined the plasmid and donor DNA and transfected into a parasite line where the known Complex IV protein *Tg*ApiCox25 had been HA tagged and contained an ATc-regulatable promoter [8]. We selected GFP positive parasites 2 days after transfection using flow cytometry and screened for positive parasites using the *Tg*ApiCox13 screen fwd and rvs primers (S2 Table). We termed the resulting parasite line 'r*Tg*ApiCox25-HA/FLAG-*Tg*ApiCox13'.

To generate a parasite line where *Tg*ApiCox25 is GFP tagged in the rHA-*Tg*ApiCox13 line, we introduced a 3' GFP tag into the locus of *Tg*ApiCox25 using an existing sgRNA-expression plasmid targeting the region around the stop codon of *Tg*ApiCox25 [8]. We amplified donor DNA encoding a tobacco etch virus (TEV) cleavage site plus a GFP epitope tag from a TEV-GFP gBlock and 50 bp flanking sequence either side of the *Tg*ApiCox25 stop codon using the *Tg*ApiCox25 tag fwd and rvs primers (S2 Table). We co-transfected the plasmid and PCR product into rHA-*Tg*ApiCox13 parasites, selected GFP positive parasites by flow cytometry 2 days post transfection and screened for successful integration using the *Tg*ApiCox25 screen fwd and rvs primers (S2 Table). We called the resulting parasite line 'rHA-*Tg*ApiCox13/*Tg*ApiCox25-GFP' and used it as the background line for complementation experiments.

To generate a constitutively expressed, wild type *Tg*ApiCox13 for complementation experiments in the rHA-*Tg*ApiCox13 line, we performed a PCR reaction on a gBlock containing *Tg*ApiCox13 and an N-terminal FLAG tag using *Tg*ApiCox13 WT complementation (comp) fwd and rvs primers (S2 Table). The PCR product was then digested with *Bgl*II and *Pst*I and ligated into equivalent sites of the pUgCTH3 vector [47]. The resulting pUgCT-FLAG-*Tg*ApiCox13 vector contained a chloramphenicol-resistance cassette, a UPRT sequence for integration in the non-essential UPRT locus of *T. gondii*, and the complementing FLAG-*Tg*ApiCox13. The vector was linearized in the UPRT flanking sequence using *Mfe*I and subsequently transfected into rHA-*Tg*ApiCox13/*Tg*ApiCox25-GFP parasites, selected on chloramphenicol, and dilution cloned. Although we suspect that integration of the plasmid will occur into the UPRT locus, we did not test this experimentally. We termed the resulting parasite line 'rHA-*Tg*ApiCox13/cFLAG-*Tg*ApiCox13$^{\text{WT}}$' (cWT).

To introduce 3' FLAG epitope tags into the loci of *Tg*ApiCox25, *Tg*Cox2a and *Tg*MPPα, we used sgRNA-expressing vectors generated previously to target the region around their stop codons [8,10]. We PCR amplified a C-terminal FLAG gBlock using gene-specific fwd and rvs tagging primers (S2 Table). We co-transfected the plasmid and PCR product into rHA-*Tg*ApiCox13 parasites, selected GFP positive parasites by flow cytometry 2 days post transfection and screened for successful integration using gene-specific fwd and rvs screening primers (S2 Table). We called the resulting parasite lines 'rHA-*Tg*ApiCox13/*Tg*ApiCox25-FLAG', 'rHA-*Tg*ApiCox13/*Tg*Cox2a-FLAG' and 'rHA-*Tg*ApiCox13/*Tg*MPPα-FLAG'.

To generate parasite lines expressing mNeonGreen (mNG) fused to full-length *Tg*ApiCox13 (mNG-*Tg*ApiCox13), the C-terminal transmembrane domain of *Tg*ApiCox13 (88-118aa, mNG-TM) or *Tg*ApiCox13 without the C-terminal transmembrane domain (1-92aa, mNG-*Tg*ApiCox13$^{\Delta\text{TM}}$), we first synthesized a synthetic DNA molecule encoding *Nco*I-mNG-

*Pst*I without a stop codon (GenScript, Jiangsu, China). We then generated a vector we termed pmNG-TgATG8 by replacing the GFP of pGFP-TgATG8 [48] with mNG, cloning into the *Nco*I and *Pst*I sites. Using cDNA of RH ΔHXGPRT as a template, we performed a PCR reaction to amplify the sequences encoding full-length *Tg*ApiCox13, the *Tg*ApiCox13 TMD, and *Tg*ApiCox13$^{\Delta TM}$ using the primer pairs mNG-*Tg*ApiCox13–fwd/mNG-*Tg*ApiCox13–rvs, mNG-*Tg*ApiCox13$^{TM}$–fwd/mNG-*Tg*ApiCox13–rvs and mNG-*Tg*ApiCox13–fwd/mNG-*Tg*ApiCox13$^{\Delta TM}$–rvs, respectively (S2 Table). Each fragment was cloned into the *Pst*I/*Pac*I sites of the pmNG-*Tg*ATG8 to replace *Tg*ATG8 using the ClonExpress II One Step Cloning Kit (Vazyme Biotech, Nanjing, China). Each construct (100 μg) was transfected into RH ΔHXGPRT parasites, selected with 25 μg/ml mycophenolic acid and 50 μg/ml xanthine and dilution cloned. We termed the parasite lines 'mNG-*Tg*ApiCox13', 'mNG-TM' and 'mNG-*Tg*ApiCox13$^{\Delta TM}$', respectively.

To generate a parasite line where *Tg*ApiCox13 is FLAG tagged in the previously generated r*Tg*NFS1 line [25], we used the *Tg*ApiCox13 CRISPR vector described above. We amplified donor DNA encoding a FLAG epitope tag and 50bp flanks of *Tg*ApiCox13 using the N-terminal FLAG gBlock and *Tg*ApiCox13 FLAG fwd and rvs primers (S2 Table). We combined the plasmid and donor DNA and transfected into r*Tg*NFS1 parasites. We selected GFP positive parasites 2 days after transfection using flow cytometry and screened for positive parasites using the *Tg*ApiCox13 screen fwd and rvs primers (S2 Table). We termed the resulting parasite line 'r*Tg*NFS1/FLAG-*Tg*ApiCox13'.

To generate several constitutively expressed Fe-S cluster binding site mutants of *Tg*Api-Cox13 (H49Q, H85Q, H49/85Q and H49/85C) to complement the rHA-*Tg*ApiCox13 line, we performed Q5 mutagenesis reactions using the pUgCT-FLAG-*Tg*ApiCox13 vector as template. Q5 mutagenesis reactions were performed according to the manufacturer's instructions (New England Biolabs) using the primer sets H49Q fwd and rvs, H85Q fwd and rvs, H49C fwd and rvs, and H85C fwd and rvs (S2 Table). We confirmed successful plasmid modification by Sanger sequencing. To generate the double H49/85Q and H49/85C mutants we first mutated the H49 position and subsequently used this as the template for the second H85 mutagenesis reaction. The vectors were linearized in the UPRT flanking sequence using *Mfe*I and subsequently transfected into rHA-*Tg*ApiCox13/*Tg*ApiCox25-GFP parasites, selected on chloramphenicol and dilution cloned. We termed the resulting parasite lines 'rHA-*Tg*ApiCox13/cFLAG-*Tg*ApiCox13$^{H49Q}$' (cH49Q), 'rHA-*Tg*ApiCox13/cFLAG-*Tg*ApiCox13$^{H85Q}$' (cH85Q), 'rHA-*Tg*ApiCox13/cFLAG-*Tg*ApiCox13$^{H49/85Q}$' (cH49/85Q) and 'rHA-*Tg*ApiCox13/cFLAG-*Tg*ApiCox13$^{H49/85C}$' (cH49/85C).

To introduce a Ty1 tag into the *Tg*SdhB locus of the rHA-*Tg*ApiCox13 and r*Tg*NFS1-HA parasite lines, we first introduced a sgRNA targeting the 3' end of the open reading frame of *Tg*SdhB into the pSAG1::Cas9-U6-UPRT vector using Q5-site directed mutagenesis according to the manufacturer's instructions (New England Biolabs). We performed the Q5 reaction using *Tg*SdhB 3' CRISPR fwd primer and the generic CRISPR rvs primer (S2 Table), confirming successful plasmid modification by Sanger sequencing. We PCR amplified a Ty1 tag with flanking sequences targeting the *Tg*SdhB locus using a Ty1 gBlock, and the primers *Tg*SdhB tag fwd and rvs (S2 Table). We co-transfected the plasmid and PCR product into rHA-*Tg*Api-Cox13 and r*Tg*NFS1-HA [25] parasites, selected GFP positive parasites by flow cytometry 2 days post transfection and screened for successful integration using *Tg*SdhB fwd and rvs screening primers (S2 Table). We called the resulting parasite lines 'rHA-*Tg*ApiCox13/*Tg*SdhB-Ty1' and 'r*Tg*NFS1-HA/*Tg*SdhB-Ty1'.

To introduce a FLAG tag into the *Tg*mtFDX locus of the rHA-*Tg*ApiCox13/*Tg*SdhB-Ty1 and r*Tg*NFS1-HA/*Tg*SdhB-Ty1 parasite lines, we first introduced a sgRNA targeting the 3' end of the open reading frame of *Tg*mtFDX into the pSAG1::Cas9-U6-UPRT vector using Q5-site

directed mutagenesis according to the manufacturer's instructions (New England Biolabs). We performed the Q5 reaction using *Tg*mtFDX 3' CRISPR fwd primer and the generic CRISPR rvs primer (S2 Table), confirming successful plasmid modification by Sanger sequencing. We PCR amplified a FLAG tag with flanking sequences targeting the *Tg*mtFDX locus using the C-terminal FLAG gBlock and the primers *Tg*mtFDX tag fwd and rvs (S2 Table). We co-transfected the plasmid and PCR product into rHA-*Tg*ApiCox13/*Tg*SdhB-Ty1 and r*Tg*NFS1-HA/*Tg*SdhB-Ty1 parasite parasites, selected GFP positive parasites by flow cytometry 2 days post transfection and screened for successful integration using *Tg*mtFDX fwd and rvs screening primers (S2 Table). We called the resulting parasite lines 'rHA-*Tg*Api-Cox13/*Tg*SdhB-Ty1/*Tg*mtFDX-FLAG' and 'r*Tg*NFS1-HA/*Tg*SdhB-Ty1/*Tg*mtFDX-FLAG'.

## SDS-PAGE, BN-PAGE and western blotting

For sodium dodecylsulfate (SDS)- polyacrylamide electrophoresis (PAGE), parasite proteins were solubilised in NuPAGE reducing sample buffer (Thermo Fisher Scientific; 1× LDS sample buffer containing 2.5% v/v 2-mercaptoethanol) and separated on a NuPAGE 12% Bis-Tris protein gel (Thermo Fisher Scientific) before transfer onto a nitrocellulose membrane. Western blotting was performed as described previously [49]. Primary antibodies used included mouse anti-FLAG (1:250–1:2,000 dilution; clone M2, Sigma catalogue number: F3165), rat anti-HA (1:500–1:2,000 dilution; clone 3F10, Sigma catalogue number 11815016001), mouse anti-GFP (1:300–1:1,000 dilution; Sigma catalogue number: 11814460001), mouse anti-Ty1 (1:2,000 dilution; clone BB2, [50]) mouse anti-mNG antibody (1:1,000 dilution; clone 32F6, Chromotek) and rabbit anti-Tom40 (1:2,000 dilution; [51]). Secondary antibodies used were horseradish peroxidase (HRP)-conjugated goat anti-mouse IgG (1:5,000–1:10,000 dilution; Abcam catalogue number: ab6789), goat anti-rabbit IgG (1:5,000 dilution; Abcam catalogue number: ab97051), HRP-conjugated goat anti-mouse IgG (1:5,000 dilution; Biosharp catalogue number: BL001A) and goat anti-rat IgG (1:10,000 dilution; Abcam catalogue number: ab97057). For probing for mouse antibodies on immunoprecipitation western blots, HRP-conjugated anti-mouse TrueBlot ULTRA antibodies (Rockland, catalogue number: 18-18817-33) were used at 1:5,000 dilution. Blots were imaged using a ChemiDoc MP imaging system (BioRad).

For Blue Native (BN)-PAGE, proteins were solubilised in Native PAGE Sample Buffer (Thermo Fisher Scientific) containing 2 mM EDTA, 1 x cOmplete protease inhibitors (Sigma) and 1% (v/v) TX-100 (Sigma). Protein samples were separated on a NativePAGE 4–16% Bis-Tris protein gel (Thermo Fisher Scientific), then transferred to a polyvinylidene difluoride (PVDF) membrane, fixed in 8% (v/v) acetic acid for 15 minutes and destained in methanol, before blocking overnight in Tris-buffered saline containing 4% (w/v) skim milk powder. PVDF membranes were probed with mouse anti-FLAG (1:250–1:2,000 dilution; clone M2, Sigma catalogue number: F3165), rat anti-HA (1:500 dilution; clone 3F10, Sigma catalogue number 11815016001), mouse anti-GFP (1:300–1:1,000 dilution; Sigma catalogue number: 11814460001), and rabbit anti-Tom40 (1:2,000 dilution; [51]). Secondary antibodies used were HRP-conjugated goat anti-mouse IgG (1: 5,000 dilution; Abcam catalogue number: ab6789), goat anti-rabbit IgG (1: 5,000 dilution; Abcam catalogue number: ab97051) and goat anti-rat IgG (1:10,000 dilution; Abcam catalogue number: ab97057).

## In-gel Complex IV assay

Proteins from parasites were solubilised in Native PAGE Sample Buffer (Thermo Fisher Scientific) containing 2 mM EDTA, 1 × cOmplete protease inhibitors (Sigma) and 1% (v/v) TX-100 (Sigma) at $2.5 \times 10^5$ parasites per μL. Samples were separated by Clear Native PAGE on a

NativePAGE 4–16% Bis-Tris protein gel (Thermo Fisher Scientific) using BN-PAGE buffers prepared without cathode buffer additive, but with sodium deoxycholate (0.05% w/v, Sigma, D6750) and *n*-dodecyl β-D-maltoside (0.01% w/v, Sigma, D4641). After separation, the lane containing the NativeMark protein ladder (Thermo Fisher Scientific) was removed and stained with Coomassie. An in-gel Complex IV activity assay was performed on the remaining lanes as described in [27]. Briefly, the gel was equilibrated in 50 mM $KH_2PO_4$ buffer (pH 7.2) for 15 minutes before incubating in freshly made Complex IV activity buffer ($KH_2PO_4$ (50 mM, pH 7.2), cytochrome *c* (0.1% w/v from horse heart, Sigma, C2506), catalase (0.1% w/v from bovine liver, Sigma, C1345), sucrose (7.5% w/v, Sigma, S9378) and 3,3′-diaminobenzidine tetrahydrochloride (DAB, 0.1% w/v, Sigma, D8001) until staining of the no ATc sample was visible. The gel was then fixed in methanol (50% v/v) and acetic acid (10% v/v), re-connected with the Coomassie-stained protein ladder and imaged using a ChemiDoc MP imaging system (BioRad).

## Sodium carbonate extraction

Sodium carbonate extractions were performed as described previously [25]. Briefly, parasite pellets were resuspended in 100 mM $Na_2CO_3$ (pH > 11) and incubated on ice for 30 min. Samples were centrifuged at low speed (1,500 ×*g*, 10 min) to remove unlysed parasites, and then at high speed (189,000 ×*g*) for 30 min at 4˚C to pellet membranes. The resulting pellet was solubilized in 1× LDS reducing sample buffer (Thermo Fisher Scientific). Proteins from the resulting supernatant were precipitated in trichloroacetic acid then solubilized in 1× LDS reducing sample buffer (Thermo Fisher Scientific).

## Co-immunoprecipitation

Immunoprecipitations were performed as previously described [51]. Briefly, r*Tg*Api-Cox25-HA/FLAG-*Tg*ApiCox13 parasites were solubilized in a 1% (v/v) TX-100-containing lysis buffer (1% v/v TX-100, 150 mM NaCl, 2 mM EDTA, 1× cOmplete protease inhibitors, 50 mM Tris pH 7.4) for an hour before being centrifuged at 21,000 ×*g* for 30 minutes at 4˚C to pellet the cell debris. The supernatant was incubated overnight at 4˚C, either with anti-HA affinity matrix (Sigma catalogue number 11815016001) or anti-FLAG M2 affinity gel (Sigma catalogue number A2220). Unbound protein samples were collected and precipitated in trichloroacetic acid. The beads were washed four times in a wash buffer (0.1% v/v TX-100, 150 mM NaCl, 2 mM ETDA, 50 mM Tris pH 7.4), and bound proteins were eluted in 1× LDS reducing sample buffer (Thermo Fisher Scientific).

## Immunofluorescence/Fluorescence microscopy assays

For immunofluorescence assays, egressed parasites were added to wells of a 6-well plate containing HFF cells growing on glass coverslips. We cultured the infected host cells for ~20 hrs before fixation in 3% (w/v) paraformaldehyde in phosphate-buffered saline (PBS; 136 mM NaCl, 10 mM phosphate, 2.7 mM KCl, pH 7.4) for 15 mins. We permeabilized fixed samples for 10 mins in 0.25% (v/v) TX-100 in PBS before blocking in 2% (w/v) bovine serum albumin in PBS overnight. Coverslips were probed with mouse anti-FLAG (1:500 dilution; clone M2, Sigma catalogue number F3165) together with rabbit anti-Tom40 ([51]; 1:2,000 dilution) primary antibodies. Secondary antibodies included goat anti-mouse AlexaFluor 546 (1:500 dilution; Thermo Fisher Scientific, catalog number A-11030), donkey anti-rabbit CF 647 (1:500 dilution; Sigma, catalog number SAB4600177), goat anti-mouse AlexaFluor 488 (1:500 dilution; Thermo Fisher Scientific, catalog number A-11029), and goat anti-rabbit AlexaFluor 546 (1:500 dilution; Thermo Fisher Scientific, catalog number A-11035). Images were acquired on

a DeltaVision Elite deconvolution microscope (GE Healthcare), and were deconvolved and adjusted linearly for contrast and brightness using SoftWoRx Suite 2.0 software.

For live-cell fluorescence imaging, parasites were transiently transfected with 30 μg of the pHsp60-RFP vector [52], which expresses RFP fused to the mitochondrial targeting peptide of *Tg*Hsp60, and then seeded onto a HFF monolayer. Forty-eight hours post-transfection, images were acquired using the Eclipse Ci-L fluorescence microscope (Nikon, Japan) equipped with a CFI Plan Fluor optical imaging system, a 100×/1.3-numerical-aperture (NA) Nikon lens, a Digital sight camera connected to PC via DS-L4 control unit and a NIS-Elements F 4.60.00 software. Adjustments for brightness and contrast were applied uniformly to the entire image.

## Plaque assays

500 parasites were added to 25 cm$^2$ tissue culture flasks containing HFF cells. Parasites were cultured in the absence or presence of ATc for seven to nine days, stained with 2% (w/v) crystal violet solution (Con and Huckers formulation; Fornine Lab Supplies or Gram's Crystal Violet solution, Sigma catalog number 94448) for at least 1 hour before washing in PBS to remove excess stain. In the case of ATc washout experiments, parasites were grown in the presence of ATc for 3 days, then the ATc was washed out and the parasites allowed to grow for a further 9 days in the absence of ATc. Control flasks grown in the absence of ATc for 8 days, or in the presence of ATc for all 12 days were also included. Flasks were imaged using a CanoScan9000F scanner.

## Seahorse XFe96 extracellular flux analysis

Seahorse XFe96 extracellular flux analyses were conducted as described previously [24]. Briefly, *T. gondii* parasites were cultured in HFF cells in the absence of ATc or the presence of ATc for 1–3 days. Parasites were mechanically egressed from the host cells through a 26 gauge needle, then filtered through a 3 μm polycarbonate filter to remove host cell debris. Parasites were washed once in base medium (Agilent Technologies) then split into two tubes: one that was resuspended in base medium supplemented containing 5 mM glucose and 1 mM gluta-mine to assess basal mitochondrial oxygen consumption rates (mOCR) of intact parasites, and one that was resuspended in base medium lacking a carbon source for 1 hour at 37°C to deplete endogenous ETC substrates, allowing subsequent assessment of the activity of individual ETC complexes. $1.5 \times 10^6$ parasites were added to wells of a Cell-Tak-coated Seahorse XFe96 cell culture plate, and adhered to the bottom by centrifugation (800 ×g, 3 min). For intact parasites, an extra 75 μL of base medium supplemented with glucose and glutamine was added to each well (final volume 175 μL). For starved parasites, the base medium was gently removed then replaced with 175 μL of mitochondrial assay solution (MAS) buffer (220 mM mannitol, 70 mM sucrose, 10 mM $KH_2PO_4$, 5 mM $MgCl_2$, 0.2% w/v fatty acid-free bovine serum albumin, 1 mM EGTA, and 2 mM HEPES-KOH pH 7.4) containing 0.002% (w/v) digitonin.

Specific substrates and inhibitors were loaded into the Seahorse XFe96 sensor cartridge ports A-D and injected into the wells during the experiments. To assess the basal mOCR of intact parasites, we loaded the following substrates into the indicated ports (with final concentration after injection indicated in parentheses): Port A, FCCP (1 μM); Port B, atovaquone (ATV; 1 μM). To assess the activity of different ETC complexes in starved, permeabilized parasites, we loaded the following substrates into the indicated ports: Port A, malate (10 mM) plus FCCP (1 μM); Port B, atovaquone (ATV; 1 μM); Port C, N,N,N',N'-tetramethyl-p-phenylene-diamine dihydrochloride (TMPD; 0.2 mM) mixed with ascorbic acid (3.3 mM); Port D, sodium azide ($NaN_3$; 10 mM). OCR and extracellular acidification rate (ECAR) measurements

were simultaneously obtained every 3 minutes for 3 repeats before and after injection of compounds. A minimum of 4 background wells were used in each plate, with and 2–3 technical replicates used for each condition. Experiments were repeated three times independently on different days with different parasite preparations.

## Analysis of *T. thermophila* Complex IV structure

The structure of *Tetrahymena* Complex IV (PDB structure 7W5Z [13]) was downloaded and evaluated in ChimeraX 1.4 [53]. The distance between two atoms in the electron transferring functional groups (heme or copper) of COX1 and COX2 and/or the Fe-S clusters of the COXFS protein was measured using the "distance" command. PNG images were exported and labelled in Adobe Illustrator.

## Data analysis

Seahorse XFe96 flux analysis data were analysed using a linear mixed-effects model to account for variation due to random effects vs fixed effects between repeats as described previously [24]. Analysis output reported the least square means, and the statistical differences between these means were tested through ANOVA (linear mixed effects) and a post hoc Tukey test. Quantification of western blotting bands was performed using ImageJ, normalised relative to the *Tg*Tom40 loading control and expressed as a percentage of the abundance on 0 days ATc. Statistical differences were tested using ANOVA followed by Tukey's multiple comparisons test or unpaired t-test with Welch's correction in GraphPad Prism.

## Supporting information

**S1 Fig. Conservation of *Tg*ApiCox13/*Hs*MiNT protein features across eukaryotic evolution.** An illustrative phylogenetic tree based on currently accepted models for eukaryotic evolution [54]. Branches were coloured based on whether the ApiCox13 homologs of clade members contain a C-terminal transmembrane domain (TMD, pink) and/or an N-terminal mitochondrial targeting peptide (mTP, green) (S1 Table). Possible points at which TMD or mTP features were gained are indicated by coloured circles.
(TIF)

**S2 Fig. Recombinant *Tg*ApiCox13 is an Fe-S cluster binding protein.** Purified recombinant *Tg*ApiCox13$^{1-93}$ and the Fe-S cluster binding site mutants *Tg*ApiCox13$^{1-93;\ H49C}$, *Tg*Api-Cox13$^{1-93;\ H85C}$, *Tg*ApiCox13$^{1-93;\ H49/85C}$, *Tg*ApiCox13$^{1-93;\ H49Q}$, *Tg*ApiCox13$^{1-93;\ H85Q}$ and *Tg*ApiCox13$^{1-93;\ H49/85Q}$ were analysed for **(A)** colour of samples (red vs clear), **(B)** molecular mass by SDS-PAGE stained with Coomassie, and for Fe-S cluster binding by **(C)** UV-visible absorption spectroscopy and **(D)** iron content measurements given as a ratio of Fe to protein in mol. Iron content measures represent the mean ± standard deviation from $n = 3$ independent experiments. One way ANOVA and a Tukey's multiple comparisons test was performed, with relevant *p*-values shown.
(TIF)

**S3 Fig. Generating FLAG tagged *Tg*ApiCox13 and ATc-regulated, HA-tagged *Tg*ApiCox13 parasite lines. (A)** Diagram depicting the 5' replacement strategy to introduce a FLAG epitope tag into the *Tg*ApiCox13 locus of r*Tg*ApiCox25-HA parasites. A sgRNA was designed to target the *T. gondii* genome near the start codon of *Tg*ApiCox13. A plasmid containing the sgRNA and GFP-tagged Cas9 endonuclease was co-transfected into *T. gondii* parasites with a PCR product encoding a FLAG epitope tag flanked by 50 bp of sequence homologous to the regions immediately up- and down-stream of the *Tg*ApiCox13 start codon. Forward and reverse

primers (*Tg*ApiCox13 screen fwd and rvs) were designed to screen parasite clones for integration of the FLAG tag at the *Tg*ApiCox13 locus, yielding a 400 bp product in the native locus and a 490 bp product in the modified locus. **(B)** PCR screening using genomic DNA extracted from putative FLAG-*Tg*ApiCox13 parasites (clones 1–4). Clones 1–3 yielded PCR products that indicated they had been successfully modified. Genomic DNA extracted from wild type (WT) parasites was used as a control. **(C)** Diagram depicting the 5' replacement strategy to simultaneously HA-tag and replace the native promoter of *Tg*ApiCox13 to generate ATc-regulated HA-*Tg*ApiCox13 parasites. The same sgRNA described in (A) was used to target the *T. gondii* genome near the start codon of *Tg*ApiCox13. A plasmid containing the sgRNA and GFP-tagged Cas9 endonuclease was co-transfected into *T. gondii* parasites with a PCR product encoding the ATc regulatable 't7s4' promoter, which contains 7 copies of the Tet operon and a Sag4 minimal promoter, plus a start codon immediately followed by a HA epitope tag, flanked by 50 bp of sequence homologous to the regions immediately up- and down-stream of the *Tg*ApiCox13 start codon. The PCR product also contains a 'spacer' region that separates the regulatable promoter from the native promoter of the *Tg*ApiCox13 gene to enable sufficient regulation. The same primers described in (A) were used to screen parasite clones for integration of the regulatable promoter and HA tag at the *Tg*ApiCox13 locus, yielding a 400 bp product in the native locus and a 2.3 kbp product in the modified locus. **(D)** PCR screening using genomic DNA extracted from putative rHA-*Tg*ApiCox13 parasites (clones 1–6). All 6 clones yielded PCR products that indicated they had been successfully modified. Genomic DNA extracted from WT parasites was used as a control.
(TIF)

**S4 Fig. General parasite metabolism is maintained upon knockdown of *Tg*ApiCox13.** Basal extracellular acidification rate (ECAR) of WT (blue), rHA-*Tg*ApiCox13 parasites (green) or rHA-*Tg*ApiCox13/cFLAG-*Tg*ApiCox13$^{WT}$ (purple) parasites grown in the absence of ATc or in the presence of ATc for 1–3 days. A linear mixed-effects model was fitted to the data and values depict the estimated marginal mean ± 95% CI of three independent experiments. ANOVA followed by Tukey's multiple pairwise comparisons test was performed, with relevant *p* values shown.
(TIF)

**S5 Fig. Parasites remain viable upon knockdown of *Tg*ApiCox13.** Plaque assays of rHA-*Tg*ApiCox13 (top), r*Tg*NFS1-HA (upper middle), r*Tg*QCR11-FLAG (lower middle) and r*Tg*ApiCox25-HA (bottom) parasites grown in the absence of ATc for 8 days (left), grown in the presence of ATc for all 12 days (middle) or pre-incubated with ATc for 3 days before washing out the ATc and growing for a further 9 days in the absence of ATc (right). Plaque assays are from a single experiment and are representative of four independent experiments. Scale bar represents 1 cm.
(TIF)

**S6 Fig. Generating FLAG tagged *Tg*Cox2a, *Tg*ApiCox25 and *Tg*MPPα in rHA-*Tg*ApiCox13 parasites.** Diagrams depict the 3' replacement strategy to FLAG-tag target genes. sgRNAs were designed to target the *T. gondii* genome near the stop codon of target genes. A plasmid containing the sgRNA and GFP-tagged Cas9 endonuclease was co-transfected into rHA-*Tg*ApiCox13 *T. gondii* parasites with a PCR product encoding a FLAG epitope tag flanked by 50 bp of sequence homologous to the regions immediately up- and down-stream of the stop codon. Genomic DNA extracted from wild type (WT) parasites was used as a control in PCRs. **(A)** Forward and reverse primers were designed to screen parasite clones for integration of the FLAG tag at the *Tg*Cox2a locus, yielding a 260 bp product in the native locus and a 361 bp

product in the modified locus. **(B)** PCR screening using genomic DNA extracted from putative *Tg*Cox2a-FLAG parasites (clones 1–8). Clones 1–6 and 8 yielded PCR products that indicated they had been successfully modified. **(C)** Forward and reverse primers were designed to screen parasite clones for integration of the FLAG tag at the *Tg*ApiCox25 locus, yielding a 385 bp product in the native locus and a 492 bp product in the modified locus. **(D)** PCR screening using genomic DNA extracted from putative *Tg*ApiCox25-FLAG parasites (clones 1–9). Clones 5 and 6 yielded PCR products that indicated they had been successfully modified. **(E)** Forward and reverse primers were designed to screen parasite clones for integration of the FLAG tag at the *Tg*MPPα locus, yielding a 274 bp product in the native locus and a 375 bp product in the modified locus. **(F)** PCR screening using genomic DNA extracted from putative *Tg*MPPα-FLAG parasites (clones 1–8). Clones 4–6 yielded PCR products that indicated they had been successfully modified.
(TIF)

**S7 Fig. Generating FLAG tagged *Tg*ApiCox13 in the ATc-regulated *Tg*NFS1 parasite line.**
**(A)** Diagram depicting the 5' replacement strategy to FLAG-tag *Tg*ApiCox13. The same plasmid described in S3A Fig containing the sgRNA and GFP-tagged Cas9 endonuclease was co-transfected into rHA-*Tg*ApiCox13 *T. gondii* parasites with a PCR product encoding a FLAG epitope tag flanked by 50 bp of sequence homologous to the regions immediately up- and down-stream of the *Tg*ApiCox13 start codon. Forward and reverse primers were designed to screen parasite clones for integration of the FLAG tag at the *Tg*ApiCox13 locus, yielding a 402 bp product in the native locus and a 492 bp product in the modified locus. **(B)** PCR screening using genomic DNA extracted from putative FLAG-*Tg*ApiCox13 parasites (clones 1–6). Clones 3, 5 and 6 yielded PCR products that indicated they had been successfully modified. Genomic DNA extracted from wild type (WT) parasites was used as a control.
(TIF)

**S8 Fig. Generating GFP tagged *Tg*ApiCox25 in rHA-*Tg*ApiCox13 parasites. (A)** Diagram depicting the 3' replacement strategy to GFP-tag *Tg*ApiCox25. The same plasmid described in S6C Fig containing the sgRNA and GFP-tagged Cas9 endonuclease was co-transfected into rHA-*Tg*ApiCox13 *T. gondii* parasites together with a PCR product encoding a GFP epitope tag flanked by 50 bp of sequence homologous to the regions immediately up- and down-stream of the *Tg*ApiCox25 stop codon. Forward and reverse primers were designed to screen parasite clones for integration of the GFP tag at the *Tg*ApiCox25 locus, yielding a 400 bp product in the native locus and a 1.2 kbp product in the modified locus. **(B)** PCR screening using genomic DNA extracted from putative *Tg*ApiCox25-GFP parasites (clones 1–6). Clones 1–3 and 5–6 yielded PCR products that indicated they had been successfully modified.
(TIF)

**S9 Fig. Generating Ty1-tagged *Tg*SdhB and FLAG-tagged *Tg*mtFDX in rHA-*Tg*ApiCox13 and *Tg*NFS1-HA parasites. (A)** Diagram depicting the 3' replacement strategy to Ty1-tag *Tg*SdhB. A plasmid containing a sgRNA targeting near the stop codon of the *Tg*SdhB open reading frame and GFP-tagged Cas9 endonuclease was co-transfected into rHA-*Tg*ApiCox13 or *Tg*NFS1-HA *T. gondii* parasites together with a PCR product encoding a Ty1 epitope tag flanked by 50 bp of sequence homologous to the regions immediately up- and down-stream of the *Tg*SdhB stop codon. Forward and reverse primers were designed to screen parasite clones for integration of the Ty1 tag at the *Tg*SdhB locus, yielding a 349 bp product in the native locus and a 477 bp product in the modified locus. **(B-C)** PCR screening to test for integration of a Ty1 tag into the *Tg*SdhB locus in **(B)** rHA-*Tg*ApiCox13 or **(C)** *Tg*NFS1-HA parasites using genomic DNA extracted from clonal parasites. Clones 1 and 3–7 in the rHA-*Tg*ApiCox13 line

**(B)** and clones 1–3 and 5–7 in the r*Tg*NFS1-HA line **(C)** yielded PCR products that indicated they had been successfully modified. **(D)** Diagram depicting the 3' replacement strategy to FLAG-tag *Tg*mtFDX. A plasmid containing a sgRNA targeting near the stop codon of the *Tg*mtFDX open reading frame and GFP-tagged Cas9 endonuclease was co-transfected into rHA-*Tg*ApiCox13/*Tg*SdhB-Ty1 or *Tg*NFS1-HA/*Tg*SdhB-Ty1 parasites together with a PCR product encoding a FLAG epitope tag flanked by 50 bp of sequence homologous to the regions immediately up- and down-stream of the *Tg*mtFDX stop codon. Forward and reverse primers were designed to screen parasite clones for integration of the FLAG tag at the *Tg*mtFDX locus, yielding a 274 bp product in the native locus and a 386 bp product in the modified locus. **(E-F)** PCR screening to test for integration of a FLAG tag into the *Tg*mtFDX locus of **(E)** rHA-*Tg*A-piCox13/*Tg*SdhB-Ty1 or **(F)** *Tg*NFS1-HA *Tg*SdhB-Ty1 parasites using genomic DNA extracted from clonal parasites. Clones 1–6 and 8 in the rHA-*Tg*ApiCox13/*Tg*SdhB-Ty1 line **(E)** and clone 3 in the *Tg*NFS1-HA *Tg*SdhB-Ty1 parasite line **(F)** yielded PCR products that indicated they had been successfully modified.
(TIF)

**S10 Fig. The structure of *Tetrahymena thermophila* Complex IV contains an Fe-S cluster protein (COXFS).** PDB structure 7W5Z [13] was evaluated in Chimera. **A)** The position of COXFS (green, the *T. thermophila* homolog of ApiCox13) in the Complex IV dimer relative to the catalytic subunits COX1 (magenta) and COX2 (orange), with all other subunits shown in grey. The heme groups of COX1 are shown in blue, and the Fe-S clusters of COXFS are shown in yellow. The inner mitochondrial membrane (IMM) is depicted by dashed lines with the intermembrane space (IMS) above and the mitochondrial matrix (MM) below. **B)** The position of COXFS (green) relative to the catalytic subunits COX1 (magenta) and COX2 (orange) without the grey subunits. The functional groups are labelled. **C)** The path and distance between the electron transferring functional groups within Complex IV are depicted by arrows. The distance between the CuB functional group and the two Fe-S clusters of COXFS are shown by dashed lines.
(TIF)

**S1 Table. Homologs of *Tg*ApiCox13 and conservation of key features.** We performed BlastP searches of the NCBI database using *Tg*ApiCox13 as a query sequence to identify homologs from other organisms, with the protein ID, percent identity to *Tg*ApiCox13, and E-value reported. We used the bioinformatics tools MitoProt II to predict presence (+, green) or absence (-, gray) of mitochondrial targeting peptides, and TMHMM to predict presence (+, magenta) or absence (-, gray) of transmembrane domains. Where appropriate, we have also provided the vEuPathDB ID.
(PDF)

**S2 Table. Sequences of primers and gBlocks used in this study.**
(PDF)

# Acknowledgments

We would like to thank Harpreet Vohra and Michael Devoy (ANU) for assistance with flow cytometry, Teresa Neeman from the ANU Statistical Consulting Unit for assistance with data analysis, and the 2019 ANU Cell Biology course for contributing to preliminary experiments to localize *Tg*ApiCox13. We are also grateful to Dr. Sébastien Besteiro (Universites de Montpellier, France) for providing the pGFP-*Tg*ATG8 construct.

## Author Contributions

**Conceptualization:** Feng Tan, Giel G. van Dooren, Jenni A. Hayward.

**Data curation:** Rachel A. Leonard, Yuan Tian, Jenni A. Hayward.

**Formal analysis:** Rachel A. Leonard, Yuan Tian, Jenni A. Hayward.

**Funding acquisition:** Feng Tan, Giel G. van Dooren.

**Investigation:** Rachel A. Leonard, Yuan Tian, Jenni A. Hayward.

**Methodology:** Rachel A. Leonard, Yuan Tian, Jenni A. Hayward.

**Project administration:** Yuan Tian, Feng Tan, Giel G. van Dooren, Jenni A. Hayward.

**Resources:** Yuan Tian, Feng Tan, Giel G. van Dooren.

**Supervision:** Feng Tan, Giel G. van Dooren, Jenni A. Hayward.

**Validation:** Rachel A. Leonard, Yuan Tian, Jenni A. Hayward.

**Visualization:** Yuan Tian, Giel G. van Dooren, Jenni A. Hayward.

**Writing – original draft:** Rachel A. Leonard, Jenni A. Hayward.

**Writing – review & editing:** Rachel A. Leonard, Yuan Tian, Feng Tan, Giel G. van Dooren, Jenni A. Hayward.

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
