## [Decision Letter · Decision Letter 0]

18 Oct 2022

Dear Ms Hayward,

Thank you very much for submitting your manuscript "An essential role for an Fe-S cluster protein in the cytochrome c oxidase complex of Toxoplasma parasites" for consideration at PLOS Pathogens. As with all papers reviewed by the journal, your manuscript was reviewed by members of the editorial board and by several independent reviewers. In light of the reviews (below this email), we would like to invite the resubmission of a significantly-revised version that takes into account the reviewers' comments.

The reviewers appreciated the work presented in this manuscript showing that TgApiCox13 is a FeS protein involved in electron transport in the mETC and important for the integrity of complex IV. The reviewers describe additional experiments that should be attempted using existing reagents to more fully understand the cellular and functional roles of TgApiCox13 including its ability to transfer FeS and the cellular consequences of depleting TgApiCox13.

We cannot make any decision about publication until we have seen the revised manuscript and your response to the reviewers' comments. Your revised manuscript is also likely to be sent to reviewers for further evaluation.

Sincerely,

Sean T Prigge

Guest Editor

PLOS Pathogens

Kami Kim

Section Editor

PLOS Pathogens

Kasturi Haldar

Editor-in-Chief

PLOS Pathogens

orcid.org/0000-0001-5065-158X

Michael Malim

Editor-in-Chief

PLOS Pathogens

orcid.org/0000-0002-7699-2064

The reviewers appreciated the work presented in this manuscript showing that TgApiCox13 is a FeS protein involved in electron transport in the mETC and important for the integrity of complex IV. The reviewers describe additional experiments that should be attempted using existing reagents to more fully understand the cellular and functional roles of TgApiCox13 including its ability to transfer FeS and the cellular consequences of depleting TgApiCox13.

Reviewer's Responses to Questions

**Part I - Summary**

Reviewer #1: In their manuscript, Leonard et al. describe the functional study of TgApiCox13, an Iron-Sulfur (FeS) protein belonging to the complex IV of the mitochondrial electron transport chain (mETC) of Toxoplasma gondii. T. gondii, as other unicellular eukaryotes, contains marked differences with the better characterized mETC of mammals, including original additional components for the complex IV, although none of these had been functionally characterized prior to this study.

Overall, the manuscript is well-written, the data presented therein are of good quality, the experimental design is sound and the conclusions are well-supported by the results obtained. This manuscript presents a very comprehensive biochemical study, demonstrating clearly that TgApiCox13 is a FeS protein involved in electron transport in the mETC and important for the integrity of complex IV.

Reviewer #2: In this manuscript, Leonard et al. have characterized a novel subunit in the Complex IV of mitochondrial electron transport chain in Toxoplasma gondii. This protein, TgCox13 or COXFS, has been predicted to be a part of Complex IV in complexome profiling studies of Plasmodium falciparum, in structural studies of Tetrahymena, and in mitochondrial proteome analysis by the authors’ own group. Bioinformatics analysis suggests TgCox13 is a small protein of 13 kDa that is homologous to a well characterized mitoNEET protein in humans, MiNT or mitoNEET3 (PMID: 29259115). In fact, humans have three mitoNEET proteins, including mitoNEET, NAF-1 and MiNT. Interestingly, MiNT has two 2Fe-2S cluster binding domains (CDGSH domains) whereas the other two mitoNEET proteins only have one. Structurally, MiNT forms a pseudosymmetric monomeric protein whereas other NEET proteins form a homodimer. TgCox13 shares a high level of sequence homology and domain topology to human MiNT, except for the C-terminal transmembrane helix. The authors have provided abundant data to show that, 1) TgCox13 binds Fe-S cluster, 2) TgCox13 is important (or essential) for parasite growth and mitochondrial respiration, 3) TgCox13 likely belongs to Complex IV, 4) TgCox13’s C-terminal transmembrane helix is important for localization, 5) TgCox13’s Fe-S cluster is likely provided by the mitochondrial iron-sulfur cluster biosynthesis pathway, 6) TgCox13’s histidine residues in the CDGSH domains are critical (especially the first histidine) for Complex IV stability and parasite survival.

Overall, Leonard et al. have provided a large amount data to investigate TgCox13’s localization, essentiality, complex formation, and critical/essential amino acid residues or domains. The manuscript is well written and easy to follow. However, there is one critical feature of TgCox13 that has not been investigated or mentioned in this study. Addressing this critical feature of TgCox13 will further clarify its role in Toxoplasma and understand how it becomes divergent from the human counterpart.

Reviewer #3: The manuscript from Leonard et al. extensively examined the role of an Fe-S cluster protein called TgApiCox13 in the cytochrome c oxidase complex in Toxoplasma, using biochemical and genetic approaches. The results clearly showed that TgApiCox13 is an Fe-S cluster binding protein critical for the integrity and activity of the cytochrome c oxidase complex. TgApiCox13 depletion leads to growth arrest of tachyzoites and the capacity to bind Fe-S cluster is critical for full activity. Given the protein composition differences between apicomplexan parasites and their animal hosts, the elucidation of the role and possible working mechanisms of the Fe-S cluster protein TgApiCox13 has important insights into the biology of electron transport chains. The manuscript was well written and easy to follow. The conclusions are well supported by the data presented.

**Part II – Major Issues: Key Experiments Required for Acceptance**

Reviewer #1: My only main criticism is that while the authors provide an extensive metabolic analysis (including the elegant use of Seahorse measurements to assess the involvement of TgApiCox13 in complex IV function), they did not analyse in details the consequences of TgApiCox13 depletion at the cellular level. As mentioned by the authors themselves (l. 545-549), mETC inhibition has been linked previously to T. gondii tachyzoites initiating a differentiation to the slow-growing bradyzoite stage, which has important implications for parasite persistence, and thus for designing anti-parasitic strategies. There are two very simple experiments that could be done to assess whether or not the TgApiCox13 mutant also follows the same trend, and which I think would bring important insights into the overall impact of TgApiCox13 depletion on parasite viability. The first one would be to do an ATc washout experiment to see if the parasites can survive TgAPiCox13 depletion to some extent: this can be simply done by performing a plaque assay and then removing the ATc from the mutant cell line after a week and monitoring the potential appearance of plaques over the course of the following week. The second experiment would be to look for the initiation of parasite differentiation upon TgApiCox13 depletion by ATc through a simple Dolichol biflorus lectin staining and microscopic observation to check for the appearance of this cyst wall marker over the course of a few days (atovaquone may be used as a positive control).

Reviewer #2: Major point

Can TgCox13 transfer Fe-S clusters to other proteins?

A common feature of human mitoNEET proteins is to transfer Fe-S clusters to other proteins (MitoNEET/PMID: 21788481, NAF-1/PMID: 24706857, MiNT/PMID: 35135884, and many other publications). Although localization of mitoNEET proteins in Toxoplasma might be different from human counterparts, the highly conserved sequence and domain features indicate that TgCox13 can potentially transfer Fe-S. Therefore, the Fe-S cluster transfer experiment is a logically must-do experiment because this will reveal TgCox13’s actual functionality. If TgCox13 can transfer Fe-S clusters, at least two hypotheses could be generated. One is TgCox13’s role in delivering Fe-S clusters to other ETC complexes such as Complex II and Complex III. Formation of super-complexes of ETC complexes supports this hypothesis. The second hypothesis is TgCox13’s role in transferring Fe-S clusters made in the mitochondrion to the cytosol. The authors have shown that TgCox13 relies on the mitochondrial iron-sulfur cluster (ISC) pathway. Therefore, the likelihood of exporting Fe-S by TgCox13 cannot be ruled out. On the other hand, if TgCox13 cannot transfer Fe-S clusters to other proteins (which will be a surprise), then it is likely that TgCox13 performs its function via oxidation-reduction of the iron and may directly transfer electrons within the Complex IV itself. To address these forementioned hypotheses, whether TgCox13 transfers Fe-S clusters must be verified.

The authors have already expressed and purified wildtype and mutant TgCox13 proteins, which will greatly facilitate the Fe-S cluster transfer experiment. As shown by the authors, TgCox13 protein is red/orange, and it shows the expected characteristic peaks in the spectrometer.

In Discussion, the authors should have a short summary of NEET proteins’ Fe-S cluster transfer ability and their roles in mitochondrial physiology in organisms studied so far (mostly model systems for sure). This will help emphasizing the unique role of this conserved protein in protozoan such as Toxoplasma.

Reviewer #3: 1. Fig 8A-C and Fig 9A. depletion of the NFS1 protein involved in Fe-S cluster synthesis reduced the protein abundance of TgApiCox13 and the authors explained this as the Fe-S binding was important to the stability of TgApiCox13. As such, when the point mutations (H49Q and/ or H85Q) were used to complement the TgApiCox13 depletion strain, the protein levels would have been different because these different mutant alleles had different Fe-S binding capacity. Such information could not be identified without a loading control in Fig 8A-C, but was available in Fig 9A. however, I am puzzled by the protein abundance patter presented here. Can the authors explain why the abundance of FLAG-TgApiCox13 of H49Q was different in parasites treated with and without ATc? And such difference was not seen with the H49Q/H85Q mutant? Similarly, why the level of HA-TgApiCox13 was higher in all three mutants than in WT complements (judging from the gel)? In addition, are the different complementation lines in Fig 8A-C isogenic (meaning that they were expressed from the same locus of the genome and driven by the same promoter)?

2. it may not be essential for this work, did the authors try to complement the TgApiCox13 depletion mutant with human MiNT or the dinoflagellate or ciliate homolog of TgApiCox13 to see whether they could complement? They may have similar functions although they localized to the mitochondria by different mechanisms (N-terminal peptide vs C-terminal TM domain).

**Part III – Minor Issues: Editorial and Data Presentation Modifications**

Reviewer #1: L. 66: correct to “Apicomplexa is a phylum comprising intracellular parasites…” or “Apicomplexans belong to a phylum of intracellular parasites…”.

Table 1: I would order this by coupling it to a phylogenetic analysis, hierarchical clustering might reveal if the C-terminal feature was inherited in related eukaryotes.

Figure 1. I think that adding an AlphaFold model of TgApiCox13, highlighting the two cluster-binding domains, the C-terminal helix, and perhaps also highlighting the mutated residues, would be particularly informative.

L. 244-248. Can it be speculated that the increase in ECAR might also reflect a compensatory mechanism in response to the decrease in mETC function?

Fig. 4D: given the extremely high error bars displayed on the graph, I doubt this can be interpreted correctly. These data are rather accessory and might be removed.

Fig. 6B, C. Overexposed immunofluorescence signals, pictures should be replaced if possible.

L. 392-393: the authors conclude that FeS biogenesis is essential for Complex IV activity as if it is a specific phenotype, but I wonder if the emphasis should be put on that as Fig. 7B shows that TgNFS1 mutants respond poorly to malate, ATV or TMPD, which is consistent with other FeS proteins being involved in complexes II and III also.

Fig. 9A, left bottom graph. I doubt that there would be no statistical significance when comparing cWT and cH49/85Q –ATc conditions: check again, or perhaps the bars are badly positioned… Also, any comment as to why there is a decrease of TgApiCox13 even in the absence of ATc?

L. 566: “a recent structural analysis of Complex IV in the ciliate Tetrahymena…”; add corresponding reference.

Reviewer #2: Minor points

1, Table 1, it is helpful to list gene IDs that are shown in vEupthDB. That will help parasitologists to search the genes.

2, Fig 2 intends to show that TgCox13 and TgCox25 belong to the same complex, as predicted by structural studies in Tetrahymena or complexome profiling in Plasmodium. The authors should also run the co-immunoprecipitation product on BN-PAGE to show that Cox13 is present in a large complex in the eluted material of Cox25 IP, and vice versa. Fig2C/2D have not provided direct evidence that TgCox13 and TgCox25 are in the same complex.

3, Fig 4B/7B. Could the authors explain why 10 mM NaN3 seems not to inhibit OCR robustly (not even as good as 1 uM atovaquone)?

4, Fig 4B, data of the complemented lines shown in 4C are missing.

5, Fig 4E, a protein loading control is missing.

6, Fig 6A, it is hard to conclude that Cox13 is enriched in the pellet based on the data shown here. Perhaps other methods of mitochondrial membrane isolation should be used to address this issue. This is critical to interrogate the role of TgCox13 because the matrix TgCox13 could be involved in other functions, not in Complex IV.

7, Fig 9A/9B, a loading control is missing for the western blots on the right panel.

8, Fig 9A, Line 468, please explain what do these up or down changes in protein abundances mean or imply.

Other points

1, Lines 38, remove the extra that

2, Line 153, H is missing in 49C.

3, Line 244-245, could the author explain why ECAR is increased?

4, Line 423, Line 426, Line 439, 84Q should be 85Q

5, Line 605, what type of French press?

6, Line 631/857/858, ug should be µg, uL should be µL

7, Lines 689/698/700/715/724/727/737 and also the main text, strains should be lines

8, Line 881, should distance be bold?

Reviewer #3: 1, table 1 is somewhat redundant with figure 1. It may be placed to the supplemental files.

2, line 271, the full name of TMPD is needed.

3, lines 423, 426 and 439 (likely elsewhere as well), should be “cH49/85Q”

4, Fig 4E, it would be helpful to indicate what exactly the readers are looking? What does that band mean?

5, line 602, expression commonly requires a lysogenic DE3 phage, so the strain is likely BL21(DE3), please check.

6, line 636, “an HA tag”

PLOS authors have the option to publish the peer review history of their article (what does this mean?). If published, this will include your full peer review and any attached files.

Reviewer #1: No

Reviewer #2: No

Reviewer #3: No
---

## [Editor Report · Decision Letter 1]

17 May 2023

Dear Ms Hayward,

We are pleased to inform you that your manuscript 'An essential role for an Fe-S cluster protein in the cytochrome c oxidase complex of Toxoplasma parasites' has been provisionally accepted for publication in PLOS Pathogens.

Best regards,

Sean T Prigge

Guest Editor

PLOS Pathogens

Kami Kim

Section Editor

PLOS Pathogens

Kasturi Haldar

Editor-in-Chief

PLOS Pathogens

orcid.org/0000-0001-5065-158X

Michael Malim

Editor-in-Chief

PLOS Pathogens

orcid.org/0000-0002-7699-2064

The authors did a good job of responding to the reviewer's comments and suggestions. New experimental results address several of the issues that the reviewers raised and alternative conclusions are discussed where appropriate.

Minor corrections:

line 93. Change 'sites' to 'site'

lines 785, 890 and 904 refer to 'open reading' when 'open reading frame' may have been intended

lines 1471 and 1484 change 'bars represent' to 'bars representing'
---

## [Editor Report · Acceptance letter]

30 May 2023

Dear Ms Hayward,

We are delighted to inform you that your manuscript, "An essential role for an Fe-S cluster protein in the cytochrome *c* oxidase complex of *Toxoplasma* parasites," has been formally accepted for publication in PLOS Pathogens.

Best regards,

Kasturi Haldar

Editor-in-Chief

PLOS Pathogens

orcid.org/0000-0001-5065-158X

Michael Malim

Editor-in-Chief

PLOS Pathogens

orcid.org/0000-0002-7699-2064